# Exploring End-to-end Differentiable Neural Charged Particle Tracking – A Loss Landscape Perspective

**Tobias Kortus**                                          *tobias.kortus@rptu.de*
*Chair for Scientific Computing*
*University of Kaiserslautern-Landau (RPTU)*

**Ralf Keidel**                                            *ralf.keidel@rptu.de*
*Chair for Scientific Computing*
*University of Kaiserslautern-Landau (RPTU)*

**Nicolas R. Gauger**                          *nicolas.gauger@scicomp.uni-kl.de*
*Chair for Scientific Computing*
*University of Kaiserslautern-Landau (RPTU)*

**On behalf of the Bergen pCT Collaboration**

**Reviewed on OpenReview:** `https://openreview.net/forum?id=1Pi2GwduEz`

## Abstract

Measurement and analysis of high energetic particles for scientific, medical or industrial applications is a complex procedure, requiring the design of sophisticated detector and data processing systems. The development of adaptive and differentiable software pipelines using a combination of conventional and machine learning algorithms is therefore getting ever more important to optimize and operate the system efficiently while maintaining end-to-end (E2E) differentiability. In this work, we lay the groundwork for E2E differentiable decision focused learning for the application of charged particle tracking using graph neural networks with combinatorial components, solving a linear assignment problem for each detector layer. We demonstrate empirically that including differentiable variations of discrete assignment operations allows for efficient network optimization, working better or on par with approaches that lack E2E differentiability. In additional studies, we dive deeper into the optimization process and provide further insights from a loss landscape perspective, providing a robust foundation for future work. We demonstrate that while both methods converge into similar performing, globally well-connected regions, they suffer under substantial predictive instability across initialization and optimization methods, which can have unpredictable consequences on the performance of downstream tasks such as image reconstruction. We also point out a dependency between the interpolation factor of the gradient estimator and the prediction stability of the model, suggesting the choice of sufficiently small values. Given the strong global connectivity of learned solutions and the excellent training performance, we argue that E2E differentiability provides, besides the general availability of gradient information, an important tool for robust particle tracking to mitigate prediction instabilities by favoring solutions that perform well on downstream tasks.

## 1 Introduction

Charged particle tracking (Strandlie & Frühwirth, 2010) is a central element in analyzing readouts generated by ionizing energy losses in particle detectors, with the goal of reconstructing distinct sets of coherent particle tracks from a discrete particle cloud measured over subsequent detector layers. Current state-of-the-art approaches in particle tracking heavily utilizes geometric deep learning in a two-phase predict-then-optimize

approach (Shlomi et al., 2021; Duarte & Vlimant, 2022; Thais et al., 2022). In an initial prediction step, relations between detector readouts described as a graph structure are modeled and quantified. Then the predicted relationships are used in a separate and disconnected step to parameterize a discrete optimization problem, generating disconnected track candidates. Separating the initial scoring from the final assignment operations, however, bares the risk of learning suboptimal solutions, that only minimize the intermediate optimization quantity of the scoring task (*prediction loss*) instead of the final *task loss* (Mandi et al., 2023). This effect is potentially further magnified by integrating particle tracking in entire component-based data processing pipelines[1] used for reconstruction and analysis, introducing additional complexities due to non-monotonic error dependencies of components on the downstream reconstruction performance, as well as complex error propagation due to entanglement of individual processing steps (Sculley et al., 2015; Nushi et al., 2017) (further discussed in Section 3.2). Providing end-to-end differentiable pipelines, where all individual components allow the propagation of gradient information, could significantly improve the overall performance of reconstruction and analysis and allow the integration of gradient-based experiment design and optimization (Dorigo et al., 2023; Aehle et al., 2023b). Providing end-to-end differentiable solutions is, however, not trivial, as the structured nature of tracks requires discrete, piecewise constant assignment operations where the gradient is undefined. While recent work, aims to bootstrap the identification of tracks to downstream tasks (Stroud et al., 2024; Reuter et al., 2024), thereby achieving end-to-end optimization, the high combinatorial complexity of finding track candidates that satisfy the imposed unique assignment constraints directly, is a major challenge. The complexity of differentiating particle tracking and reconstruction pipelines in general is further examined and discussed in Aehle et al. (2023b). The *decision-focused learning* or *predict-and-optimize* paradigm (Kotary et al., 2021; Mandi et al., 2023) provides a general framework where prediction of intermediate scoring and optimization for structured outputs are directly integrated in the training procedure by embedding the combinatorial solver as a component of the network architecture, allowing to optimize the main objective in an E2E manner. This framework has already been proven highly efficient for various fields of applications such as natural language processing, computer vision, or planning and scheduling tasks (Mandi et al., 2023). In this work, we propose and explore a predict-and-optimize framework for charged particle tracking, providing, to our knowledge, the first fully differentiable charged particle tracking pipeline for high-energy physics[2]. Further, we aim to provide first insights into the importance and usability of end-to-end particle tracking for fully differentiable reconstruction and analysis pipelines analyzing the loss landscape (e.g., sharpness/flatness of minima, connectivity of local minima, and similarity of learned models) of the tracking network. We choose this simplification over an analysis of an entire pipeline to reduce the overall complexity, introduced by other components, obscuring the influence of E2E tracking on the overall optimization performance. Further, E2E differentiability is still an active area of research for high-energy physics, with differentiability yet to be provided or improved for some components, making a meaningful and robust analysis infeasible. Our main contributions in this paper summarize as follows, serving as a foundation for further work on E2E charged particle tracking:

1. We propose an edge-classification architecture for particle tracking, that operates on a filtered line graph representation of an original hit graph, providing a strong inductive bias for the dominant effect of particle scattering.

2. We then formalize the generation of track candidates as a layer-wise linear sum assignment problem, for which we provide gradient information based on previous work by Vlastelica et al. (2020).

3. We demonstrate the competitive performance of our proposed network architecture for both traditional and E2E optimization, comparing it with existing tracking algorithms for the *Bergen pCT* detector prototype.

4. We find significant predictive instabilities between the different training paradigms as well as across random initializations, arguing the importance of E2E differentiability for robust particle tracking,

---

[1]Processing pipelines for, e.g., Bergen pCT project and the ALICE, ATLAS and CMS experiments, depicting the complexity and variety of individual chained components, are described in detail in Aehle et al. (2023b); Buncic et al. (2015); Åkesson et al. (2005); Lange & the CMSCollaboration) (2011).

[2]The source code and data utilized in this study are detailed in Appendix F, including all relevant source code, datasets, and results necessary to reproduce the findings presented in this paper.

providing necessary tools to constrain the solution space of tracking models to a subset optimizing an additional metric of the downstream tasks.

5. Further, we reveal, analyzing the local and global structure of the loss landscape, strong connectivity between the trained representations both for E2E training and optimization of a prediction loss. Together with the high predictive instability, we argue that the strong connectivity of diverse reconstruction policies enables efficient end-to-end optimization in reconstruction and analysis pipelines, favoring tracking solutions that provide beneficial performance on downstream tasks.

6. Finally, we point out a coupling between the interpolation parameter defined by the blackbox gradients and the prediction stability, suggesting the choice of sufficiently small values for optimization.

## 2 Related Work

Charged particle tracking is a well-studied problem, with the majority of advancements originating from developments in high-energy physics research at particle facilities (e.g., CERN). For the last decades, tracking algorithms have been dominated by conventional model-based pattern recognition and optimization algorithms such as Kalman filter (Frühwirth, 1987), Hough transform (Hough, 1959), or cellular automaton (Glazov et al., 1993). Here we specifically want to point out existing work by Pusztaszeri et al. (1996) related to our work, using combinatorial optimization on handcrafted features for particle tracking providing solutions that satisfy unique combination of vertex detector observations using a five-dimensional assignment model. With the ever-accelerating advancement and availability of deep learning algorithms, coupled with the increasing complexity and density of collision events, a significant demand has been imposed on novel reconstruction algorithms, resulting in faster and more precise models. While initial studies mainly relied on basic regression- and classification models based on *convolutional neural networks (CNN)* or *recurrent neural networks (RNN)* (Farrell et al., 2017a;b), remarkable progress has been made recently by computationally efficient and well-performing network architectures based on *geometric deep learning (GDL)* (Shlomi et al., 2021; Thais et al., 2022) leveraging sparse representations of particle events in combination with *graph neural networks (GNN)*. Here, the vast majority of current designs can be categorized into one of two main schemes. *Edge classification tracking* (Farrell et al., 2018; Ju et al., 2020; Duarte & Vlimant, 2020; Baranov et al., 2019; Heintz et al., 2020; DeZoort et al., 2021; Elabd et al., 2022; Murnane et al., 2023) predicts for each graph edge connecting two particle hits in subsequent layers a continuous output score, which is then used to construct track candidates. In *object condensation tracking* (Kieseler, 2020; Qasim et al., 2022; Lieret & DeZoort, 2023; Lieret et al., 2023), GNN's are trained with a multi-objective object condensation loss. In contrast to scoring edges in the graph, object condensation embeds particle hits in an N-dimensional cluster space, where hits belonging to the same track are attracted to a close proximity while hits from different tracks are repelled from each other. While both approaches provide great tracking performances, they require a separate optimization step (e.g., clustering) to generate final track candidates in a disconnected step, thus lacking an end-to-end differentiable architecture, preventing gradient flow throughout the whole reconstruction process. Recent progress has been made using reinforcement learning (Kortus et al., 2023) for charged particle tracking, allowing to differentiate through the discrete assignment operation using variants of the policy gradient or score function estimator approach, maximizing an objective function $J(\theta)$, defined as the expected future reward obtained by the agent's policy. Additional work by e.g., Stroud et al. (2024); Reuter et al. (2024) proposed the fusion of tracking with track-fitting procedures (track parameter estimation), avoiding the optimization over discrete assignment operations.

## 3 Theory and Background

In this work, we focus on reconstructing particle tracks generated in the pixelated *proton computed tomography* (pCT) detector prototype, proposed by the *Bergen pCT Collaboration* (Alme et al., 2020; Aehle et al., 2023a). This detector is designed for generating high-resolution computer tomographic images using accelerated protons and is composed using a total of 4644 high-resolution ALPIDE CMOS Pixel sensors (Mager, 2016; Aglieri Rinella, 2017) developed for the upgrade of the Inner Tracking System (ITS) of the ALICE experiment at CERN. To capture the residual energy and path of the particle, the detector is arranged in a

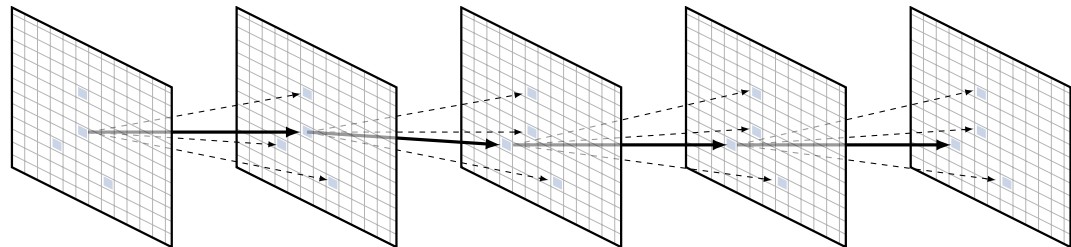

Figure 1: Schematic representation of the combinatorial complexity of reconstructing particle tracks over multiple subsequent layers. Marked in bold lines is the correct particle track. Marked in dashed lines are only possible next track segments originating from the previous correct segment. During reconstruction, all possible combinations of segments over all hits are considered.

multi-layer structure, aligned in two *tracking* and 41 *calorimeter* layers. Each of the tracking layers, used for capturing the incoming particle path with as little interaction as possible, is separated from each subsequent layer by a 55.18 mm and 39.58 mm air gap respectively. The following calorimeter layers are separated from each other with a 3.5 mm aluminum slate. A detailed description of the detector geometry can be found in Alme et al. (2020). In this detector setup, multiple simultaneous particle trajectories, generated by a 230 MeV scanning *pencil beam*, are captured distal to a patient as discrete readouts of pixel clusters in the sensitive layers which have to be reconstructed into particle tracks under the influence of physical interactions with the detector material, randomly deflecting the particle from its straight path (ref. Appendix C). With the high particle density in the detector, the resulting combinatorial explosion of track candidates, where each hit combination in subsequent layers need to be considered as possible partial solutions, requires efficient and reliable algorithms that are able to simultaneously reconstruct disjoint particle tracks by uniquely connecting particle hits over multiple layers in the detector (ref. Figure 1).

### 3.1 Particle Interactions in Matter

The path of protons at energies relevant for pCT is mainly influenced by interactions of the particles with both electrons and atomic nuclei changing the path of the accelerated particle (Gottschalk, 2018; Groom & Klein, 2000), posing additional difficulties in recovering the traversal path of the particle throughout a multi-layer particle detector. In the following, we briefly describe all three relevant effects in increasing order regarding their influence on the particle trajectories:

1. **Ionizing energy loss**: When interacting with negatively charged electrons in the atom's outer shell, protons lose a fraction of their initial energy, due to attracting forces between the particles (Bloch, 1933), causing the particle to stop after a traversal range, proportional to the particle's energy. Due to the relatively small mass of the electron compared to the high momentum of the accelerated proton, the proton remains on its original path.

2. **Elastic nuclear interactions**: However, when repeatedly interacting elastically with the heavier atomic nuclei, the proton is deflected from its original path due to the repelling forces between nuclei and particle. Integrated over multiple interactions, this effect, also referred to as *Multiple-Coulomb scattering* (MCS), follows an approximately Gaussian shape proportional to the particle's energy as well as the amount and density of material traversed (Gottschalk, 2018).

3. **Inelastic nuclear interactions**: Finally, on rare occasions, particles suffer from inelastic interactions with an atomic nucleus. In this case, the proton undergoes a destructive process, where the primary can knock out one or multiple secondary particles of various types from the target nucleus. Due to the stochastic nature of the event, the outgoing path of particles is highly complex in its nature and involves significantly larger angles compared to MCS (Gottschalk, 2018). Inelastic interactions are therefore extremely difficult to reconstruct while losing any meaningful information for image reconstruction due to its stochastic behavior.

## 3.2 Component-based Processing Pipelines

Modern data processing pipelines are increasingly powered by data-driven or machine learning algorithms, requiring the efficient interplay of multiple individually optimized tasks that are strongly coupled and entangled with each other (Sculley et al., 2015; Nushi et al., 2017). The coupling of data-driven algorithms introduces additional challenges that need to be considered in order to ensure operability and maintainability while providing reproducible and optimal performance Sculley et al. (2015). While the influences of component-based processing pipelines utilizing machine learning are manyfold, we focus in the following on the following aspects, that influence the initial design and operation, especially benefiting from end-to-end optimizable/differentiable solutions:

- **Stochastic nature of optimization:** In contrast to conventional model- or rule-based algorithms, data-driven algorithms are sensitive to random mechanisms, such as parameter initialization, stochastic optimization, or parallel operations on GPUs, reducing the overall stability of the system.

- **Error propagation:** While individual model performances are often not only affected by prediction instabilities (Lopez-Martinez et al., 2022), the different sources of errors propagate throughout the entire pipeline, resulting in unpredictable error patterns in the final output of the pipeline due to the existent complex entanglements of the downstream processing steps.

- **Non-monotonic error:** Finally, due to the entanglement of the individual components, model improvements of individual components are not necessarily bound to an improvement of the overall system performance Nushi et al. (2017). While this seems unintuitive at first, this behavior can be caused by downstream tasks being optimized on the erroneous output signal of the upstream part of the pipeline or misalignment of the individual components' optimization metric and overall system performance.

To tackle the challenges associated with component-based pipelines, different solutions have been developed for mitigating the effect. This includes, for example, the development of modular processing pipelines, aiming to decouple the individual pipeline components to reduce the complex effects of entanglements Modi et al. (2023). Increasing effort has also been put into the development of E2E-differentiable or in general optimizable ML pipelines (Yu et al., 2022; Hilprecht et al., 2023), allowing to optimize the entirety of components while optimizing the final output of interest. A similar trend, utilizing end-to-end differentiable solutions, can also be found in high-energy physics (Dorigo et al., 2023; Aehle et al., 2023b).

## 3.3 Loss Landscape Analysis

Optimization of neural networks in high-dimensional parametric spaces is notoriously difficult to visualize and understand. While theoretical analysis is complex and usually requires multiple restrictive assumptions, recent work addressed this problem by providing empirical tools for loss landscape analysis providing compressed representations for characterizing and comparing different architectures or optimization schemes. We provide in the following section a brief introduction to a subset of methods, used for the analyses in the later sections. Here, we closely follow the methodology and taxonomy provided by Yang et al. (2021) helping us to characterize and assess differences and qualities of traditional and end-to-end differentiable particle tracking with graph neural networks by characterizing and comparing the general form of the loss landscape as well as the global connectivity of the landscape.

**Loss surfaces:** To obtain a general understanding of the smoothness and shape of the loss landscape Li et al. (2018) proposed a method to visualize two-dimensional loss surfaces along random slices of the loss landscape centered on the estimated parameters $\theta^*$ according to

$$f(\alpha, \beta) = \mathcal{L}(\boldsymbol{\theta}^* + \alpha\boldsymbol{\nu} + \beta\boldsymbol{\eta}). \tag{1}$$

Here, $\boldsymbol{\nu}$ and $\boldsymbol{\eta}$ are random vectors, spanning a 2D-slice through the high-dimensional loss landscape. While this technique is not fully descriptive and only a partial view of the optimization landscape, Li et al. (2018)

demonstrates that the qualitative characteristic and behavior of the loss landscape is consistent across different randomly selected directions. In our analysis, we use the first two eigenvectors with the highest eigenvalues of the hessian matrix (Chatzimichailidis et al., 2019) of the trained network parameters $\boldsymbol{\theta}^*$, providing the loss surface along the steepest curvature.

**Mode connectivity:** Garipov et al. (2018) and Draxler et al. (2018), demonstrated in independent work the existence of non-linear low-energy connecting curves between neural network parameters. While this approach was initially proposed for generating ensembles, Gotmare et al. (2018) and Yang et al. (2021) demonstrated the usefulness of this approach for comparing different initialization and optimization strategies as well as characterizing global loss landscape characteristics. For finding connecting curves of form $\psi_\theta(t) : [0, 1] \to \mathbb{R}^d$, Garipov et al. (2018) defines an optimization schemes using two modes $\hat{w}_a$, $\hat{w}_b$ (weights after training), minimizing the integral loss alongside the parameterized curve approximated by minimizing the expectation of randomly sampled points, following a uniform distribution:

$$\mathcal{L}(\theta) = \int_0^1 \mathcal{L}(\psi_\theta(t))dt = \mathbb{E}_{t \sim U(0,1)}\left[\mathcal{L}\left(\psi_\theta(t)\right)\right]. \tag{2}$$

For quantifying the connectivity of minima generated using two-step and E2E optimization, we follow the recommended parametrization of the learnable curve used in Yang et al. (2021), defined by a Bézier curve with three anchor points ($\hat{w}_a$, $\hat{w}_b$ and a learnable anchor $\theta$), with

$$\psi_\theta(t) = (1-t)^2\hat{\boldsymbol{w}}_a + 2t(1-t)\boldsymbol{\theta} + t^2\hat{\boldsymbol{w}}_b. \tag{3}$$

We additionally evaluate the linear connectivity both as a baseline and a measure for mechanistic similarities (Neyshabur et al., 2020; Juneja et al., 2022; Lubana et al., 2023).

**Representational and functional similarities:** Analyzing network similarities in both representations and outputs is an effective tool in comparing results of network configurations, providing an estimate of proximity in the loss landscape, especially for globally well-connected minima (Yang et al., 2021).

Achieving high network similarity is especially desirable for our use cases, as differences in reconstructed tracks might influence results of the downstream tasks, such as image reconstruction for pCT or statistical analysis for HEP experiments.

To gain an insight into the behavior of the trained networks, we thus quantify both the similarity of learned representations using CKA similarities (Kornblith et al., 2019) and prediction instability using the min-max normalized disagreement (Klabunde et al., 2023).

- **Linear-CKA** Linear *central kernel alignment* (CKA) is widely used to quantify a correlation-like similarity metric (Kornblith et al., 2019) to compare learned representations of neural network layers of same or different models. CKA compares representations via the Hilbert-Schmidt Independence Criterion (HSIC) according to

$$\text{CKA}(\boldsymbol{K}, \boldsymbol{L}) = \frac{\text{HSIC}(\boldsymbol{K}, \boldsymbol{L})}{\sqrt{\text{HSIC}(\boldsymbol{L}, \boldsymbol{L})\text{HSIC}(\boldsymbol{K}, \boldsymbol{K})}}. \tag{4}$$

  where $\boldsymbol{K} = \boldsymbol{X}\boldsymbol{X}^T$ and $\boldsymbol{L} = \boldsymbol{Y}\boldsymbol{Y}^T$ are the gram matrices calculated for the representations of two layers with $\boldsymbol{X} \in \mathbb{R}^{n \times d_1}$ and $\boldsymbol{Y} \in \mathbb{R}^{n \times d_2}$. To cope with the large number of processed nodes and edges processed, we use the batched Linear-CKA leveraging an unbiased estimator of the HSIC over a set of minibatches (Nguyen et al., 2020).

- **Prediction instability** Prediction instability or churn captures the average ratio of disagreement between predictions of different models $f_1$ and $f_2$ (Fard et al., 2016; Klabunde & Lemmerich, 2023). For a general multi-class classifier, the disagreement is defined as following

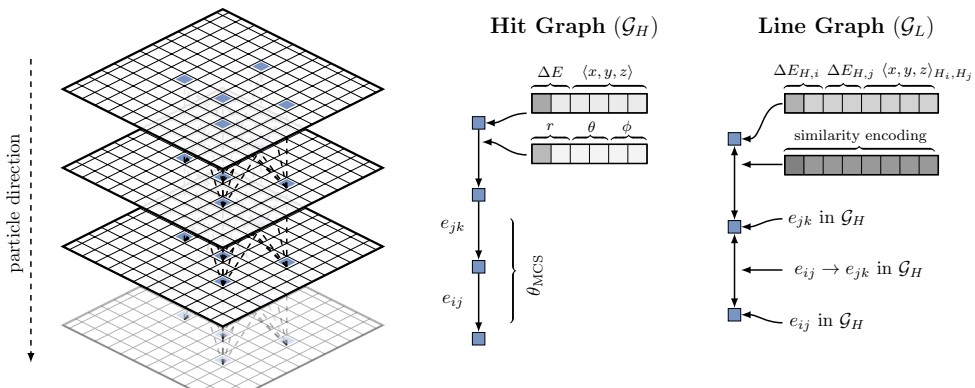

Figure 2: Schematic representation of directed hit graph $(\mathcal{G}_H)$ and undirected line graph $(\mathcal{G}_L := L(\mathcal{G}_H))$ using detector readout data simulated for the Bergen pCT prototype detector over multiple detector layers.

$$d(f_1, f_2) = \mathbb{E}_{x, f_1, f_2} 1 \left\{ \arg\max f_1(\boldsymbol{x}) \neq \arg\max f_2(\boldsymbol{x}) \right\}. \tag{5}$$

This concept naturally translates to binary classifiers as used in this paper, where the argmax is replaced by a threshold function. This metric, however, is difficult to interpret since the value range is bounded by

$$|q_{Err}(f_1) - q_{Err}(f_2)| \leq d(f_1, f_2) \leq \min(q_{Err}(f_1) + q_{Err}(f_2), 1), \tag{6}$$

where $q_{Err}$ is the error rate of the model $f_1$ or $f_2$, respectively (Klabunde & Lemmerich, 2023). We thus use the min-max normalized disagreement (Klabunde & Lemmerich, 2023), defined as

$$d_{\mathrm{norm}}(f_1, f_2) = \frac{d(f_1, f_2) - \min d(f_1, f_2)}{\max d(f_1, f_2) - \min d(f_1, f_2)}. \tag{7}$$

with $\min d(f_1, f_2) = |q_{Err}(f_1) - q_{Err}(f_2)|$ and $\max d(f_1, f_2) = \min(q_{Err}(f_1) + q_{Err}(f_2), 1)$ giving us similarities in a bounded interval of $[0, 1]$.

## 4 Predict-then-track and Predict-and-track Framework

In this section, we introduce our end-to-end differentiable *predict-and-track* (PAT) and two-step *predict-then-track* (PTT) approaches, including preprocessing steps directly aimed at the specific data gathered during proton computed tomography. We closely follow existing state-of-the-art edge-classifying approaches, with modifications to match the algorithm to the specific challenges of reconstruction in a digital tracking calorimeter, to provide a generally applicable blueprint and transferrable results that can be easily adapted to different detector geometries and data structures.

### 4.1 Hit Graph and Edge (Line) Graph Construction

In this work, we parameterize the detector data as an undirected *line graph* $\mathcal{G}_L = (\mathcal{V}_L, \mathcal{E}_L)$ generated from an initially generated *hit graph* $\mathcal{G}_H = (\mathcal{V}_H, \mathcal{E}_H)$ containing all the detector readouts of a single readout frame. In an initial hit graph structure, we represent all particle hit centroids in a readout frame as a set of graph vertices $v_{H,i}$ and describe possible partial track connections (segments) between two consecutive layers in the detector by edge connections $e_{H,ij}$. This representation is widely used for particle tracking, as it allows describing and parameterizing the particle hits and possible partial solutions in a single sparse structure. Yet, to capture a richer representation of the scattering behavior over combinations of two consecutive segments, providing a strong inductive bias for modelling the most likely path of the particle in the detector, under the

influence of elastic nuclear interactions, we transform this initial representation into a *line graph* ($\mathcal{G}_L$), where each edge in $\mathcal{G}_H$ is transformed into a node in $G_L$ (ref. Figure 2). Further, edges are generated between nodes $v_{L,i}, v_{L,j}$ that share a common vertex in $\mathcal{G}_H$, constructing descriptions of track deflections of candidates over three consecutive layers. This transformed representation allows to efficiently aggregate and compare information of scattering behavior, providing important information on possible track segments in one edge feature, that can be aggregated by a single GNN message, as supposed to the more complex aggregation in $\mathcal{G}_H$ over a two-hop neighborhood (ref. Figure 2). To provide machine-readable and differentiable features (w.r.t. simulation output) we parameterize both vertex ($\boldsymbol{v}_{L,i}$) and edge ($\boldsymbol{e}_{L,ij}$) features[3] of the line-graph representation as

$$
\begin{aligned}
\boldsymbol{v}_{L,i} &= (\Delta E_{H,i} \| \Delta E_{H,j} \| \langle x,y,z \rangle_{H,i} \| \langle x,y,z \rangle_{H,j}) \\
\boldsymbol{e}_{L,ij} &= \begin{cases} \sin(\omega \cdot [1 - \sqrt{s_{\cos}}]) & \text{if } i = 2k \\ \cos(\omega \cdot [1 - \sqrt{s_{\cos}}]) & \text{if } i = 2k+1 \end{cases},
\end{aligned}
\tag{8}
$$

where $\Delta E_H$ is the deposited energy in the sensitive layers, and $\langle x,y,z \rangle_H$ is the global position of the particle hit in the detector[4] for the adjacent nodes $i$ and $j$. Further, each edge is parameterized by a positional encoding, similar to Vaswani et al. (2017). However, instead of using discrete token positions, we leverage cosine similarities $s_{\cos}$ between the directional vectors of two track segments in $\mathcal{G}_H$ as proposed by Kortus et al. (2023), injecting additional information on the relative likelihood of observing a specific transition dynamic of track candidates over segment combinations over three consecutive detector layers under the underlying physical interaction mechanisms. The importance and usefulness of the line-graph transformation, especially its beneficial parameterization of segment pairs, functioning as a strong inductive bias by condensing information on particle scattering into a single feature, is analyzed and confirmed in detail in Appendix D. Here, we highlight the performance gain compared to a conventional hit-graph baseline (De-Zoort et al., 2021), parameterized according to the description in (Kortus et al., 2023). Additional results demonstrating the importance of segment information, provided by the positional encoding mechanism, are presented in (Kortus et al., 2023).

To reduce the complexity of the graph and to minimize the combinatorial explosion of edges in $\mathcal{G}_L$, we remove implausible edges with high angles in $\mathcal{G}_H$, measured orthogonal to the sensitive layer. We find suitable independent thresholds for both calorimeter and tracking layer edges as a trade-off between reducing the graph size and minimizing the number of removed true edges required for constructing the particle tracks (see Appendix A). Both thresholds are determined on the train-set and are kept constant for evaluation. Despite the elimination of graph edges, using a line-graph representation still introduces computational overhead for reconstruction. Additional results quantifying the impact of the line-graph representation are presented in Appendix D.

## 4.2 Edge Scoring Architecture

Following the basic idea of track reconstruction using an edge classification scheme (Heintz et al., 2020; DeZoort et al., 2021), we compose a graph neural network based on the interaction network (IN) architecture (Battaglia et al., 2016; 2018) as proposed in (DeZoort et al., 2021), predicting edge scores capturing the likelihood of every possible track segment candidate in $\mathcal{G}_H$. However, we use a slightly modified architecture to predict node scores on the line graph representation $\mathcal{G}_L$, which directly correspond to the edge scores in $\mathcal{G}_H$ (see Figure 3). Following is the formulation of edge and node updates in a generic message passing formulation, as well as the final scoring function based on node representation on $\mathcal{G}_L$. In a first step, updated edge representations are calculated using a concatenated vector containing the edge attributes, describing

---

[3]In the following, unless otherwise specified, we use a simplified notation for $\boldsymbol{v}_{L,i}$ and $\boldsymbol{e}_{L,ij}$, where $\boldsymbol{v}_i$ and $\boldsymbol{e}_{ij}$ defaults to to edge- and node-features of the line graph representation $\mathcal{G}_L$. The additional identifyer $L$ is omitted in favor of readability.

[4]We use the continuous energy deposition and position outputs of the Monte-Carlo simulations here to be able to calculate gradients w.r.t. position and energy deposition. The parameters can be replaced with the cluster size and cluster centroid position accordingly. However, this representation requires additional work to provide differentiable solutions for pixel clustering.

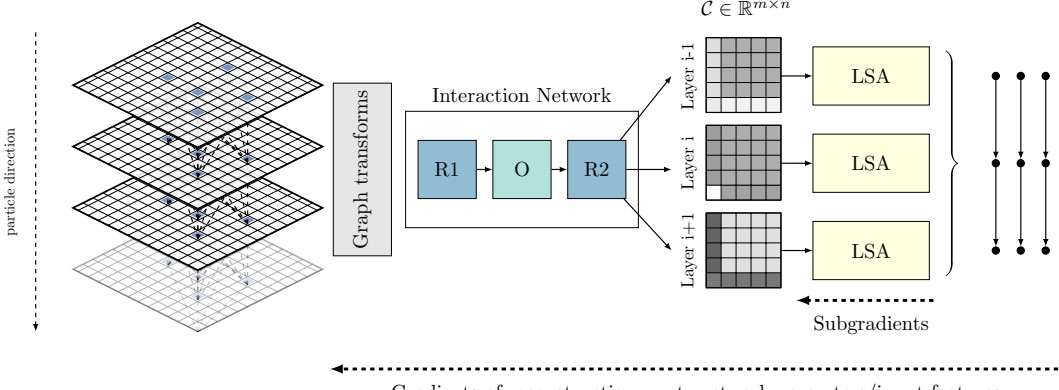

Figure 3: Combination of interaction network style architecture and layerwise combinatorial assignment for particle tracking, providing gradient information using linear interpolations of the optimization mapping.

the scattering behavior of the triplet-candidate, together with the node attributes $\boldsymbol{v}_i^{(0)}$ and $\boldsymbol{v}_j^{(0)}$ of adjacent nodes according to

$$e_{ij}^{(1)} = \phi_{R,1}\left(\boldsymbol{v}_i^{(0)}, \boldsymbol{v}_j^{(0)}, e_{ij}^{(0)}\right). \tag{9}$$

Here $\phi_{R,1}$ is a multilayer feed-forward (FF) network mapping the concatenated input to a fixed size representation with $\phi_{R,1} : \mathbb{R}^{2d_{\text{node}}+d_{\text{edge}}} \to \mathbb{R}^{d_{\text{hidden}}}$.

In a subsequent step, all node features of $\mathcal{G}_L$ are updated in an aggregation step using the current node representation and edge information aggregated from the direct neighborhood of the respective node $v_i$ according to

$$\boldsymbol{v}_i^{(1)} = \phi_O\left(\boldsymbol{v}_i^{(0)}, \max_{j \in \mathcal{N}_i}\left(e_{ij}^{(1)}\right)\right), \tag{10}$$

and transformed using a learned mapping, parameterized by a feed-forward network, with $\phi_O : \mathbb{R}^{d_{hidden}} \to \mathbb{R}^{d_{hidden}}$. Given the dynamic number of incoming edges for each graph vertex, either due to the changing numbers of hit readouts in subsequent layers of $\mathcal{G}_H$ or the required generalization ability to different particle densities of the proton beam, we use a node-agnostic max-aggregation scheme, providing empirically better generalization abilities compared to non-node-agnostic aggregation schemes (Joshi et al., 2021).

To predict for each edge in $\mathcal{G}_H$ an edge score, we perform a final reasoning step, transforming the node embeddings of each node in $\mathcal{G}_L$ into a single scalar output using a feed-forward network $\phi_{R,2}$:

$$s_{ij} = \sigma\left(\phi_{R,2}\left(\boldsymbol{v}_i^{(1)}\right)\right) \tag{11}$$

We generate the final edge scores $s_{ij} \in [0, 1]$ using a sigmoid activation function as proposed in Heintz et al. (2020); DeZoort et al. (2021). We augment the multi-layer feed-forward networks using batch normalization (Ioffe & Szegedy, 2015), to mitigate the risk of vanishing gradients and smoothen the optimization landscape (Santurkar et al., 2018), providing better training and inference performance.

### 4.3 Track Construction as a Differentiable Assignment Problem

Generating discrete track candidates from the continuous edge-score predictions requires the application of a discrete (non-differentiable) algorithm that matches the tracks with the lowest total cost. Optimally,

this would require solving a multidimensional assignment problem (MAP) to minimize the total cost of all assignments over the entire detector geometry. This exact definition is, however, impractical in real-world usage since solving a MAP is NP-hard (Pierskalla, 1968), requiring approximate solutions. In the related literature, different optimization procedures, such as DBSCAN, union-find or partitioning algorithms, are used frequently (DeZoort et al., 2021; Ju et al., 2021; Lieret & DeZoort, 2023) independently of the training procedure, to construct disconnected tracks during inference. In this work, we rely on a relaxation of the original MAP formulation to a layer-wise linear sum assignment problem (LSAP), finding a solution $y(\mathcal{C})$ in form of discrete assignments that minimizes

$$
y(\mathcal{C}): \quad \underset{y \in \mathcal{Y}}{\arg\min} \quad \sum_{(i,j) \in \mathcal{E}} y_{ij} c_{ij}
$$
$$
\text{s.t.} \quad \sum_{i \in \mathcal{V}_S} y_{ij} = 1, \quad j \in \mathcal{V}_T
$$
$$
\sum_{j \in \mathcal{V}_T} y_{ij} \le 1, \quad i \in \mathcal{V}_S \tag{12}
$$
$$
y_{ij} \in \{0, 1\}, \quad i, j \in \mathcal{V}_T \times \mathcal{V}_S
$$

for every detector layer. Here, $c_{ij} \in \mathcal{C}$ is the assignment cost for the edge combination $i, j$, where $i \in \mathcal{V}_S$ is a node from the set of source nodes, defined by all particle hits in the detector layer $l$, and $j \in \mathcal{V}_T$ is a node from the set of target nodes in the adjacent detector layer $l-1$ respectively. Further, $y_{ij}$ is the respective assignment policy defining whether the edge should be assigned under the reconstruction policy. We provide, as described below, additional gradient information w.r.t. $\mathcal{C}$ through a linearized representation of the solver mapping (Vlastelica et al., 2020), enabling the efficient optimization of the combined tracking algorithm, directly minimizing the assignment error. While the results in Section 5.1 demonstrate good performance for the choice of the LSA assignment, we want to note here that an increasing amount of detector noise and particle scattering can significantly influence the performance due to the required assignment without any notion of noise.

**Edge-costs and solver:** For the application of charged particle tracking we define a dense assignment cost matrix $\mathcal{C} \in \mathbb{R}^{|\mathcal{V}_F| \times |\mathcal{V}_T|}$, where each entry with an existing edge in $\mathcal{G}_H$ is set as $1 - s_{ij}$ [5]. For all other entries, we assign the cost matrix an infinite cost, marking this assignment as infeasible:

$$
\mathcal{C} = \begin{cases} 1 - s_{ij} & \text{if } e_{ij} \in \mathcal{E}_H \\ \infty & \text{if } e_{ij} \notin \mathcal{E}_H \end{cases}, \tag{13}
$$

The dense cost matrix allows us to solve the assignment problem reasonably efficient in $\mathcal{O}(n^3)$ using the Jonker-Volgenant algorithm (LAPJV) (Jonker & Volgenant, 1987). Similarly, for larger tracking detectors with a sparser occupation per area and thus connectivity, this notation can be replaced by a sparse cost matrix and solved with the sparse LAPMOD variant of the Jonker-Volgenant algorithm (Volgenant, 1996) respectively.

**Blackbox differentiation:** For this general formulation of LSAP a wide range of work of predict-and-optimize schemes providing an end-to-end optimization ability to combinatorial optimization problems exist based on e.g., the interpolation of optimization mappings (Vlastelica et al., 2020; Sahoo et al., 2023), continuous relaxations (Amos & Kolter, 2017; Elmachtoub & Grigas, 2017; Wilder et al., 2019) or methods that bypass the calculation of gradients for the optimizer entirely using surrogate losses (Mulamba et al., 2021; Shah et al., 2022). For an in-depth review and comparison of existing approaches, the mindful reader is referred to Geng et al. (2023). Geng et al. (2023) demonstrated for all major and representative types of

---

[5]Equivalently, we could directly predict the edge cost $c_{ij}$ by the graph neural network instead of using the edge score $s_{ij}$. However, we chose this particular notation to stay compatible with the output of the predict-then-optimize variant.

end-to-end training mechanisms for bipartite matching problems, similar regret bounds. We thus base our selection of a technique solely on simplicity and generalizability of the approach to other optimizers and use-cases. Vlastelica et al. (2020) defines a general blackbox differentiation scheme for combinatorial solvers of form $y(\boldsymbol{C}) = \arg\min_{y \in \mathcal{Y}} c(\boldsymbol{C}, y)$ by considering the linearization of the solver mapping at the point $y(\hat{\boldsymbol{C}})$ according to

$$\nabla_{\boldsymbol{C}} f_\lambda(\hat{\boldsymbol{C}}) := -\frac{1}{\lambda} \left[ y(\hat{\boldsymbol{C}}) - y_\lambda(\boldsymbol{C}') \right], \quad \text{where} \quad \boldsymbol{C}' = \text{clip}\left( \hat{\boldsymbol{C}} + \lambda \frac{dL}{dy} \left( y(\hat{\boldsymbol{C}}) \right), 0, \infty \right). \tag{14}$$

Here, $y(\hat{\boldsymbol{C}})$ and $y_\lambda(\boldsymbol{C}')$ are standard and perturbed-cost output of the combinatorial solver, $\lambda$ is a hyperparameter controlling the interpolation and $dL/dy$ is the gradient of the reconstruction loss w.r.t. the output of the combinatorial solver. Using a linearized view of the optimization mapping around the input allows remaining with exact solvers without the necessity of using any relaxation of the combinatorial problem.

**Cost margins:** The usage of discrete assignment does not provide any confidence scores, thus task losses of correctly assigned edges are always exactly zero, even for marginal differences between cost elements, potentially having a noticeable impact on the generalization performance (Rolínek et al., 2020b;a). To enforce larger margins between predicted costs, Rolínek et al. (2020b;a) introduced ground-truth-induced margins where a negative or positive penalty is added on the predicted costs based on the ground truth labels. Later, Sahoo et al. (2023) introduced noise-induced margins, removing the previous ground-truth dependence by adding random noise to the predicted costs. However, for the large number of edges in our use case of particle tracking, we found both methods to be highly unstable, even for small margin factors.

### 4.4 Network Optimization

While all previous building blocks are shared between the proposed predict-and-track and predict-then-track framework, the main difference lies in the optimization procedure of the network. We aim to optimize the prediction quality of both networks by reducing a specific loss function minimizing the disagreement between prediction and ground truth, which is defined as a binary label for every edge in $\mathcal{G}_H$. We treat splitting particle tracks containing secondary particles as noisy labels due to the relatively low production rate and generally short lifetime of most secondary particles (e.g., electrons). This avoids the need for computationally costly tracing algorithms that follow the particle generation of the Monte-Carlo (MC) simulation for the creation of the training data and avoids generating assignment rules for complex particle interactions. Given the true edge labels, we minimize for the predict-and-optimize scheme the *hamming loss* of the assignments over a batch of $N$ randomly sampled hit graphs according to

$$\mathcal{L}(\{y_n\}_{1:N}, \{f(\mathcal{G}_n)\}_{1:N}) = \frac{1}{N} \sum_{n=1}^{N} \frac{1}{|\mathcal{E}_n|} \sum_{(i,j) \in \mathcal{E}_n} w_{ij} \cdot (y_{ij} \cdot (1 - \hat{y}_{ij}) + (1 - y_{ij}) \cdot \hat{y}_{ij}), \tag{15}$$

where $\hat{y}_{ij}$ is the assignment of the $k$-th edge in hit graph $\mathcal{G}_H$ and $y_{ij}$ is the ground truth assignment of the track. We perform additional weighting of the reconstruction loss for tracking and calorimeter layer to account for the different scattering behavior. Similarly, we use the *binary-cross entropy (BCE) loss* of the predicted edge scores for the predict-then-optimize version for comparison. To reduce the overall memory footprint of the optimization, we implement the gradient calculation of a single batch using gradient accumulation as N individual predictions over a single graph. We use for all following studies an RMSProp optimizer (Hinton et al., 2012) with a learning rate of $1e-3$, which we found to be significantly more stable than Adam (Kingma & Ba, 2015) and more robust to the selection of the learning rate compared to standard SGD. We train all network variants PAT and PTT for 10,000 training iterations, where in each iteration a random minibatch of N graphs is sampled from the training set. In initial experiments, we observed that the E2E architectures showed the tendency to get unstable after reaching a certain reconstruction performance. We thus perform selective "early stopping", where we used the training checkpoint (evaluated every 100 iterations) with the best validation performance, determined based on the purity of the reconstructed tracks.

# 5 Experimental Results and Analysis

**Dataset:** For the studies reported in this work, we rely on MC simulation of detector readout data, which we generate using GATE (Jan et al., 2004; 2011) version 9.2 based on Geant4 (version 11.0.0) (Agostinelli et al., 2003; Allison et al., 2006; 2016). We generate different datasets for training (100,000 particles) and validation (5,000 particles), using a 230 MeV pencil beam as a beam source with water phantoms of various thicknesses between particle beam and detector. For comparability purposes, the test set, containing 10,000 particles in total, is taken from Kortus et al. (2022). A detailed listing of all data sources is provided in Appendix F. We generate for the training and validation simulations hit- and line graphs with 100 primaries per frame ($p^+/F$), according to the description in Section 4.1, which we consider the expected target density for the constructed detector. Further, we generate hit graphs for the test set with 50, 100 and 150 $p^+/F$ to assess the generalization ability to unseen densities.

**Hyperparameter settings:** We share model hyperparameters for both PAT and PTT framework, documented in detail in Appendix B. As we consider the interpolation parameter $\lambda$ as an important tunable parameter, potentially impacting the optimization behavior, we analyze the effect of the parameter choice on the tracking performance. For this study, we select values of 25, 50 and 75 covering a range that is well inside the specified value ranges defined for similar applications (Vlastelica et al., 2020; Rolínek et al., 2020b).

**Track reconstruction and filtering:** During inference, particle tracks are constructed combining the trained reconstruction networks (PTT and PAT) in all configurations with the linear assignment solver, creating unique assignments of particle hits to tracks[6]. To remove particle tracks produced by secondary particles or involving inelastic nuclear interactions, we apply a track filtering scheme similar to Pettersen et al. (2020) and Kortus et al. (2023) removing implausible tracks after reconstruction based on physical thresholds. In this work, we limit the track filters to an energy deposition threshold (625 keV in the last reconstructed layer), ensuring the existence of a Bragg peak in the reconstructed track.

**Baseline models:** As baseline models, providing reference values for quantifying the track reconstruction performance of the proposed architectures, we select both a traditional iterative track follower algorithm (Pettersen et al., 2020)[7], and a reinforcement learning (RL) based tracking algorithm (Kortus et al., 2023). The first baseline finds suitable tracks by iteratively searching for track candidates minimizing the total scattering angle. Similarly, the RL-based algorithm aims to learn a reconstruction policy functioning as a learned heuristic to the algorithm by (Pettersen et al., 2020) by iteratively maximizing the expected likelihood of observing the scattering behavior determined by MCS during training. Similar to our line-graph parameterization in Section 4.1, Kortus et al. (2023) augment the simpler hit-graph features using segment pair information as an inductive bias for tracking. Yet, due to the sequential nature of RL, the pairwise information is queried only for the current partial track candidates without explicitly building the full line-graph (see Kortus et al. (2023) for more details).

**Quantification of model performances:** To compare the quality of the reconstruction, we quantify the performance based on true positive rate (TPR), false positive rate (FPR) of the assigned edges as well as reconstruction purity ($p$) and efficiency ($\epsilon$) of entire tracks, defined respectively as

$$TPR = \frac{n_{TP}}{n_{TP} + n_{FN}}, \quad FPR = \frac{n_{FP}}{n_{FP} + n_{TN}}, \quad p = \frac{N_{rec,+}^{filt}}{N_{rec,+/-}^{filt}}, \quad \epsilon = \frac{N_{rec,+}^{filt}}{N_{total}^{prim}}. \tag{16}$$

Here $n_{TP}$, $n_{TN}$, $n_{FN}$, and $n_{FP}$ are the number of true-positive, true-negative, false-negative and false-positive reconstructed edge segments. Further, $N_{rec,+}^{filt}$ is the number of correctly reconstructed particle tracks after filtering, $N_{rec,+/-}^{filt}$ is the total number of reconstructed tracks after applying the track filter, and $N_{total}^{prim}$ is the total number of primary tracks (without inelastic nuclear interactions) in a readout frame. To

---

[6]Visualizations of a selection of reconstructed tracks can be found in Appendix C

[7]The source code for the track follower proposed by Pettersen et al. is provided as a software component in the Digital Tracking Calorimeter Toolkit (https://github.com/HelgeEgil/DigitalTrackingCalorimeterToolkit)

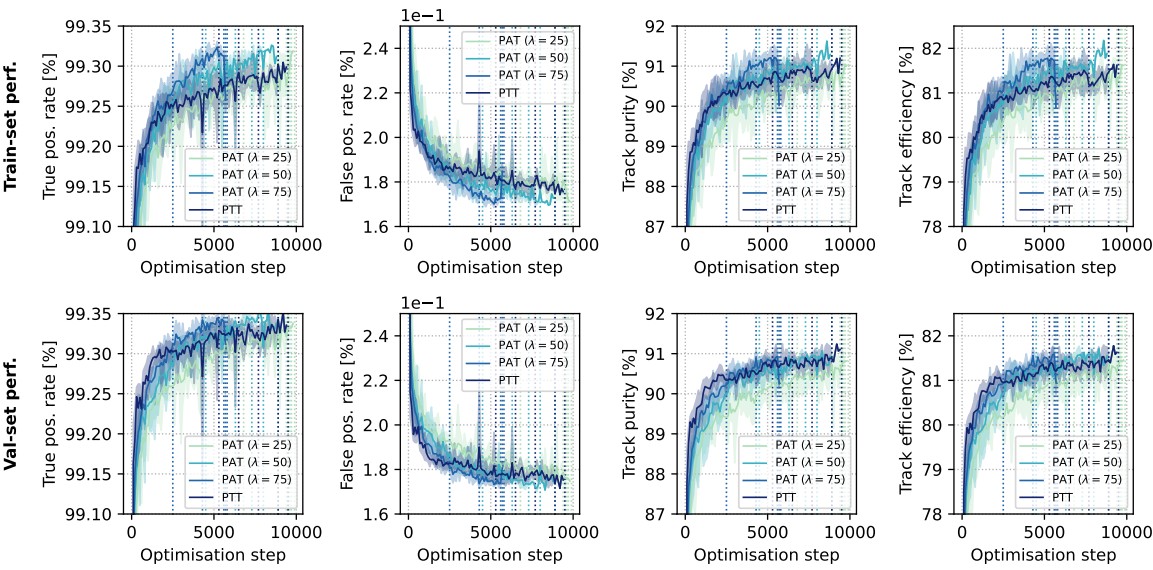

Figure 4: True positive and false positive rate of assignments together with purity and efficiency of track reconstruction, evaluated for predict-and-track and predict-then-track as a function of training steps.

provide confidence intervals for the model performance as well as the following analysis of the loss landscapes, we calculate the results over five independently trained networks with different random initializations.

## 5.1 Training and Inference Performance

Figure 4 visualizes the intermediate model performances for every 100 training iterations, determined by true-positive rate, false-positive rate, purity and efficiency, both on the training and validation dataset. Additionally, marked with horizontal dotted lines is the iteration of each model run with the highest validation purity. Noticeably, all model instances (PAT and PTT) display similar training performances with a significant improvement of all metrics over the first 1000-2000 training iterations. Continuing from there, the end-to-end trained networks, especially with a lambda of 50 and 75, shows slightly improved performance compared to the model minimizing the prediction loss. However, this performance advantage does not translate to the validation dataset, where once again all models show almost identical results. This lack of generalization ability can be likely traced back to the absence of cost margins, as described in Section 4.3, which we had to remove due to the amount of instability introduced by this mechanism. The replacement of this mechanism with an equivalent, more stable mechanism is thus likely beneficial.

Further, Figure 4 demonstrates that the end-to-end differentiable versions with an interpolation factor $\lambda$ of 50 and 75 can converge faster than the PTT variant. PAT ($\lambda = 50$) and ($\lambda = 75$) converges on average in $7000 \pm 1512$ and $4780 \pm 1263$ training steps, respectively, while PTT requires on average $7580 \pm 1536$ steps to converge. In contrast, PAT ($\lambda = 25$) requires with $9080 \pm 1151$ more steps than PPT.

Additional performance results are provided in Table 5.1. Here, the performance of the trained models is compared for different phantom and particle density configurations. In addition, the performance results of a traditional track follower procedure, which was developed for the particular use-case of particle tracking in the DTC prototype, are provided as a baseline.

Similar to the results provided in Figure 4, all model configurations, trained on the task- and the prediction-loss, demonstrate near identical reconstruction purity and efficiency over all tested phantom and particle configurations. Analogous to Vlastelica et al. (2020), we find that the exact choice of $\lambda$ is not critical for the reconstruction quality and only impacts the trainings- and convergence speed with larger $\lambda$ (ref. Figure 4). Further, the graph neural networks outperform the baseline algorithm in almost all performance metrics over

Table 1: Reconstruction performance (purity $p$ and efficiency $\epsilon$) for 100-200 mm water phantoms and 50-150 primaries per frame $p^+/F$. Results marked with **\*** use a layer-wise reconstruction scheme, with the initial transition generated using ground truth.($\uparrow$: higher is better; $\downarrow$ lower is better)

| $p^+/F$ | Algorithm | 100 mm Water Phantom | | 150 mm Water Phantom | | 200 mm Water Phantom | |
|---|---|---|---|---|---|---|---|
| | | $p$ [%] ($\uparrow$) | $\epsilon$ [%] ($\uparrow$) | $p$ [%] ($\uparrow$) | $\epsilon$ [%] ($\uparrow$) | $p$ [%] ($\uparrow$) | $\epsilon$ [%] ($\uparrow$) |
| 50 | Track follower Pettersen et al. (2020) | 87.1±0.0 | 78.1±0.0 | 89.4±0.0 | 81.1±0.0 | 90.8±0.0 | 82.1±0.0 |
| | RL-based Kortus et al. (2023)**\*** | 92.5±0.2 | 81.5±0.3 | 93.7±0.2 | 84.0±0.4 | 94.4±0.2 | 85.4±0.4 |
| | GNN$_{PTT}$ (BCE. edge score) | 94.9±0.1 | 83.8±0.0 | 96.2±0.2 | 86.9±0.2 | 96.4±0.0 | 88.7±0.1 |
| | GNN$_{PAT}$ ($\lambda = 25.0$) | 94.9±0.1 | 83.8±0.1 | 96.2±0.1 | 87.0±0.1 | 96.5±0.1 | 88.9±0.1 |
| | GNN$_{PAT}$ ($\lambda = 50.0$) | 95.0±0.2 | 83.9±0.2 | 96.3±0.2 | 87.1±0.2 | 96.5±0.1 | 89.0±0.1 |
| | GNN$_{PAT}$ ($\lambda = 75.0$) | 95.0±0.2 | 83.9±0.2 | 96.3±0.1 | 87.1±0.1 | 96.5±0.0 | 89.0±0.0 |
| 100 | Track follower Pettersen et al. (2020) | 80.6±0.0 | 71.7±0.0 | 84.7±0.0 | 76.4±0.0 | 85.8±0.0 | 77.5±0.0 |
| | RL-based Kortus et al. (2023)**\*** | 85.6±0.3 | 75.2±0.5 | 88.8±0.5 | 79.0±0.5 | 89.5±0.4 | 80.8±0.5 |
| | GNN$_{PTT}$ (BCE. edge score) | 87.4±0.3 | 75.2±0.2 | 91.9±0.3 | 82.4±0.3 | 91.7±0.2 | 83.8±0.1 |
| | GNN$_{PAT}$ ($\lambda = 25.0$) | 87.3±0.2 | 75.0±0.2 | 91.7±0.2 | 82.1±0.3 | 92.2±0.2 | 84.4±0.2 |
| | GNN$_{PAT}$ ($\lambda = 50.0$) | 87.4±0.3 | 75.1±0.2 | 92.0±0.1 | 82.4±0.2 | 92.5±0.1 | 84.6±0.1 |
| | GNN$_{PAT}$ ($\lambda = 75.0$) | 87.4±0.2 | 75.1±0.2 | 91.9±0.1 | 82.4±0.1 | 92.3±0.2 | 84.4±0.2 |
| 150 | Track follower Pettersen et al. (2020) | 75.6±0.0 | 67.2±0.0 | 80.1±0.1 | 72.2±0.0 | 82.5±0.0 | 74.6±0.0 |
| | RL-based Kortus et al. (2023)**\*** | 80.5±0.4 | 70.8±0.6 | 83.8±0.7 | 74.4±0.6 | 85.3±0.6 | 76.9±0.5 |
| | GNN$_{PTT}$ (BCE. edge score) | 77.5±0.4 | 65.0±0.3 | 84.9±0.3 | 75.3±0.3 | 87.2±0.2 | 79.6±0.2 |
| | GNN$_{PAT}$ ($\lambda = 25.0$) | 76.7±0.2 | 64.3±0.2 | 84.8±0.3 | 75.1±0.3 | 87.6±0.3 | 80.1±0.3 |
| | GNN$_{PAT}$ ($\lambda = 50.0$) | 76.8±0.3 | 64.4±0.3 | 85.1±0.2 | 75.5±0.2 | 88.1±0.4 | 80.6±0.4 |
| | GNN$_{PAT}$ ($\lambda = 75.0$) | 76.6±0.1 | 64.3±0.1 | 85.0±0.2 | 75.4±0.2 | 87.8±0.2 | 80.4±0.2 |

all configurations, with only the configuration of 150 $p^+/F$ and 100 mm water phantom as a slight outlier in this tendency. Here, the track follower and RL-based tracker demonstrate a higher reconstruction efficiency, while the reconstruction purity is still higher for the GNN architectures.

## 5.2 Evaluation of Local and Global Loss Landscapes

To analyze and characterize the training and inference behavior of the end-to-end and two-step tracking approaches and gain an understanding of the effects of end-to-end optimization and its hyperparameters, as well as an intuition regarding the usability and effectiveness of the approach in an E2E-differentiable processing pipeline (as discussed in Section 3.2), we visualize (Li et al., 2018) and characterize the loss landscape structure, closely following the methodology and taxonomy by Yang et al. (2021). We specifically focus on the global connectivity of the loss landscape, as it allows us to infer conclusions about the optimization of particle tracking in an isolated setting as well as an integrated operation in a processing pipeline, also considering gradient information from downstream tasks. We further augment the evaluation by additional analysis of functional similarities (Klabunde et al., 2023) of the optimized models, allowing us to quantify the risk of predictive instabilities of the reconstruction network as well as potential opportunities for E2E fine-tuning, gained by diverse but connected minima. All used algorithms are detailed in Section 3.3. Additional implementation details are listed in Appendix E.

### 5.2.1 Local Structure of the Task Loss Surfaces for Decision-focused Learning

Figure 5 visualizes the two-dimensional loss surfaces (Hamming loss) of the first four runs of all configurations in terms of filled contour maps in logarithmic scale. Noticeably, all loss surfaces show similar and consistent shape and patterns in the local 2D loss landscape around the found minima, both over both random initializations and choices of the interpolation parameter $\lambda$. Projected onto two dimensions, the surfaces show a mostly convex structure with wide and flat regions along the minima, supporting the previous findings of good training performance of the E2E training configuration. This pattern also often coincides with good generalization performance (Chaudhari et al., 2016). However, Dinh et al. (2017) demonstrates that the notion of flatness is not sufficient to reason about the generalization ability itself. We, argue that in this case, the large flat regions likely are conditioned and defined by the range of sensitivity of predictions along various parameter configurations. We further strengthen this intuition by comparing the hamming loss with the BCE loss surfaces in Section 5.2.2, demonstrating that the hamming loss surface strongly correlates with

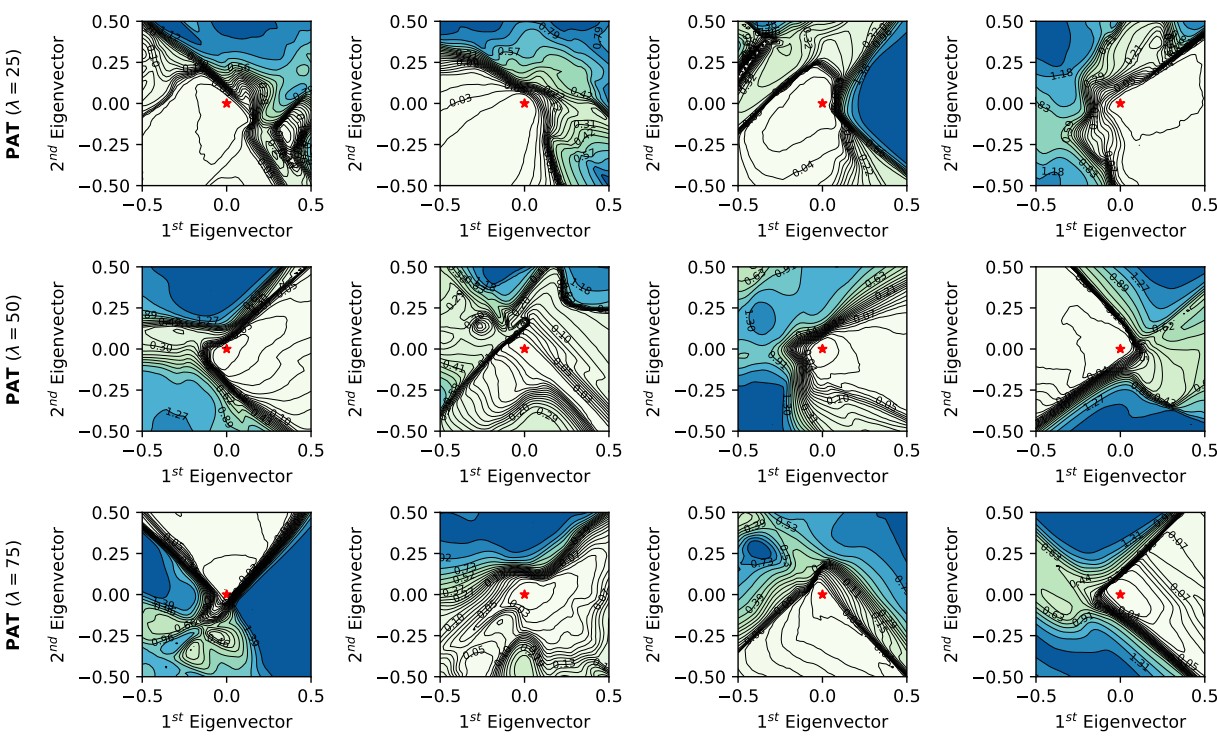

Figure 5: Two-dimensional loss surfaces of PAT framework in logarithmic scale with annotated contour lines. Marked with ★ are the trained network parameters.

a flattened version of the BCE-loss. While a significant amount of the projected loss surface is occupied by a flat, low-loss area, all loss surfaces show a significant loss barrier near the trained network parameters, demonstrating the convergence of the networks to outputs close to decision boundaries (see. Section 4.3). Providing an alternative to cost margins, discussed in Section 4.3, may thus be helpful to improve the generalization performance of the network.

### 5.2.2 Agreement of Prediction Loss and Task Loss Surfaces

While optimizing prediction and task loss use different loss functions on both intermediate and final outputs that generally do not coincide Mandi et al. (2023), Figure 4 and Table 5.1 revealed substantial similarities both during training and inference. Figure 5 indicates that the similarities directly translate, and thus are most likely caused by the similar shape of the projected loss landscape of prediction and task loss. For all tracking networks, trained minimizing the prediction loss, we find similar shapes and pattern for both the prediction loss (BCE) and task loss surface (Hamming). We especially emphasize the existence of minima in the loss surface with strongly correlated shapes, demonstrating the best agreement around the minimum itself [8]. While the overall shape of the loss surface suggests generally good agreement between the two losses, the surfaces also show significant differences in relative loss for suboptimal solutions. While this did not seem to be significant for our work, theoretical work indicates increasingly growing regrets for models optimizing the prediction loss if the parametric model class is ill-posed and there is little data available (Hu et al., 2022; Elmachtoub et al., 2023). For well-specified models, experimental results reported in Geng et al. (2023) further strengthen our findings that regret for decision-focused learning strategies in matching problems closely line up with the ones obtained with two-step training, minimizing the prediction loss.

---

[8]The difference of prediction loss and task loss vanishes for perfect predictions of the two-step approach with perfect confidences (either exactly zero or one).

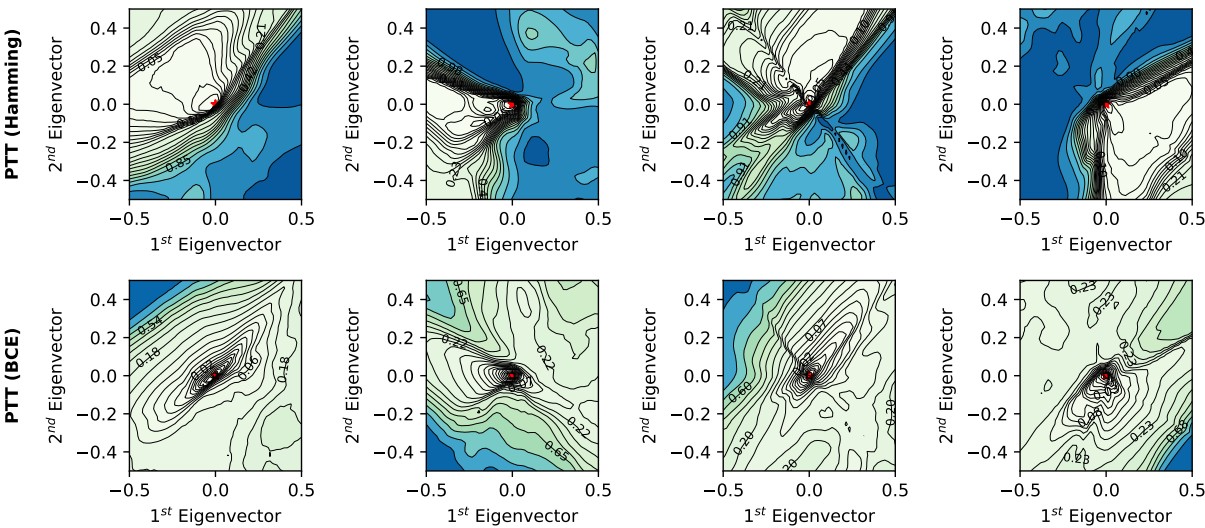

Figure 6: Visualization and comparison of task (top) and prediction loss (bottom) for trained models generated using models optimized with the predict-then-track (PTT) framework in logarithmic scale with annotated contour lines. Marked with ★ are the trained network parameters.

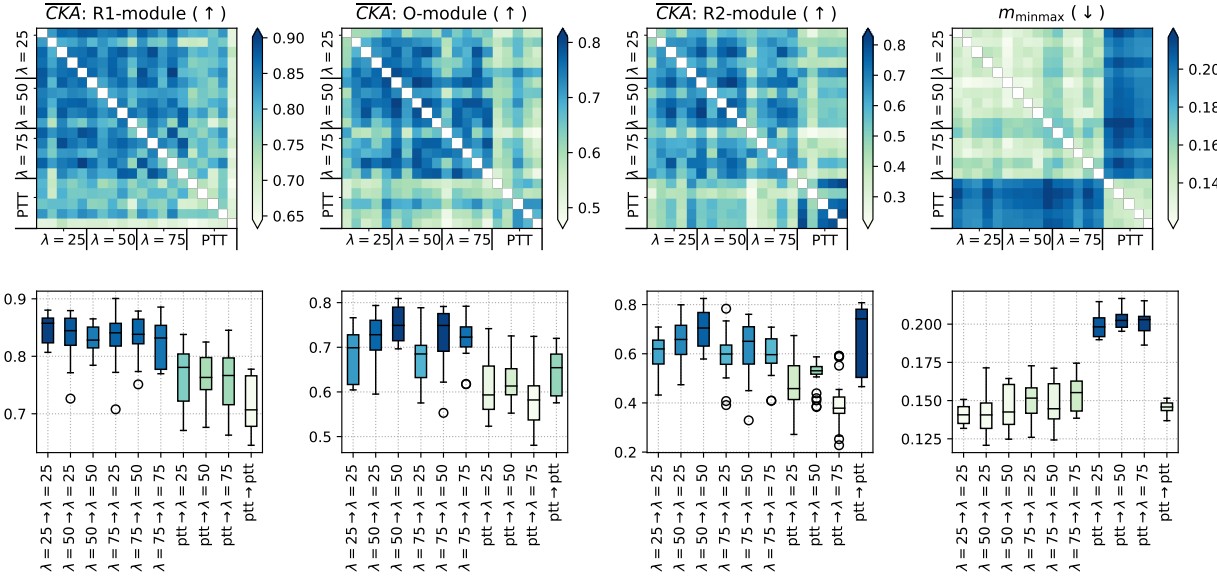

Figure 7: Representational and functional similarities, expressed in terms of CKA-similarities and min-max normalized disagreement, measured for all combinations of training configuration and random initialization (top row) with compressed representation aggregated over all random initializations (bottom row) (↑: higher is better; ↓ lower is better)

## 5.3 Representational and Functional Similarities

The similar training and inference performance (Section 5.1) as well as the strong correlation of prediction and task loss surfaces (Section 5.2) indicate strong connections between the E2E and two-step training

paradigms. However, previous results lack information about parameter and prediction level similarities and differences, thus providing only limited information of the connectivity in the global loss landscape. We thus continue analyzing the inherent similarities of the networks to get a more profound understanding of existing structural differences in both training paradigms.

Figure 7 visualizes the representational similarities (measured as average CKA similarities between layer representations) and functional similarities (measured as min-max normalized disagreement in predictions). Here, the first row shows all elementwise similarities with an empty diagonal, and the second row shows group wise similarities across random initializations.

Noticeably, PAT-based models reveal consistently moderate to high functional similarities in all network modules. Especially the first two network layers, representing the message passing and aggregation steps of the GNN architecture, show significantly higher similarities compared to their two-step counterpart. This effect is consistent both across random initialization and different combinations of interpolation factors $\lambda$. The representational similarities in the final network module are comparable for both decision-focused and two-step learning, despite the two-step models exhibiting lower similarities in the preceding layers.

Overall, combinations of PTT and PAT demonstrate the lowest representational similarities, indicating that both E2E and two-step training can find similar performing network parameters, that, however, show significantly larger differences in their internal representation compared to models trained using the same optimization paradigm.

While the average reconstruction performance across multiple readout frames is unaltered by changes in the learned network representations (ref. Table 5.1), subtle prediction instability caused by different learned prediction mechanisms can have a noticeable impact on downstream tasks. Especially for medical applications such as pCT, robustness and stability of the classifier is essential to assure reproducibility and avoid unpredictable changes in the following downstream tasks. To avoid comparing the outcome of downstream tasks, which is computationally expensive and depends on multiple intermediate steps, we analyze functional similarities or prediction instability of the tracking as a helpful proxy, complementing the previous results.

As illustrated in Figure 7, both PTT and PAT-based models show substantial functional disagreement in a similar range, with median disagreement rates around 15%. We want to specifically point out the increase in disagreement for models with higher interpolation values (see Appendix G for the statistical significance analysis), suggesting choosing smaller $\lambda$ values for improved stability of the model. Further, the experiments demonstrate, similar to the representational similarity results, a significant increase in disagreement for combinations containing both PAT and PTT variants. Here, we can observe median disagreement rates of approximately 20%, corresponding to an increase in prediction instability of approximately 33%.

**Interpretation:** *We find that the substantial degree of prediction instability and different levels of similarity in the learned representations pose a considerable risk, potentially influencing the overall performance, effectiveness, and reliability of entire component-based processing pipelines (see Section 3.2) in the application domain. While we do not expect significant effects on, e.g., the average reconstruction results of pCT, especially for homogeneous materials, the prediction instability could have a measurable impact on the contrast and spatial resolution. The instability in downstream reconstruction remains unquantified and is left for future work to maintain the conciseness of this initial study. By propagating information from a downstream task, E2E optimization provides an important tool to restrict the solution set and thus control the quality of learned solutions. While some of the instabilities can be avoided to improve reproducibility, other sources of randomness introduced by processing hardware and implementation, e.g., nondeterministic scatter operation of PyTorch Geometric (Klabunde & Lemmerich, 2023) still remain and display a noteworthy source of uncertainty and unreliability, propagated throughout the entire processing pipeline.*

### 5.4 Global Connectivity of Local Minima

Given the findings of Section 5.3, demonstrating substantial disagreements in model predictions, we now aim to quantify the global shape and connectivity of the loss landscape. Table 2 and Table 3 summarize the mode connectivity, comparing connecting Bézier curves between model parameters as specified in Section 5.2. To get a more profound understanding of the global landscape for prediction and task loss, we generate curves

using both the BCE-loss on the predicted edge-scores and the Hamming loss on the assigned edges. Here, we take advantage of our findings in Section 5.2.1, suggesting a strong agreement between Hamming-loss and BCE-loss for low-loss values. In addition, we provide the linear connectivity between two models as a baseline. Further, Neyshabur et al. (2020); Juneja et al. (2022) and Lubana et al. (2023) demonstrate in their work that trained models deficient in linear connectivity, are mechanistic dissimilar[9]. This allows us to gain further insides on the stability of the trained models from a loss landscape perspective, supporting the results in Section 5.3.

Table 2: Comparison of minimum, maximum, and average hamming loss and true positive rate for models sampled alongside low-energy connecting linear and Bézier curve (three anchor points; minimizing hamming or binary-cross-entropy loss) between two modes of same model type. (↑: higher is better; ↓ lower is better)

| | | Hamming loss [$\times 1e^{-3}$] (↓) | | | True positive rate [%] (↑) | | |
|---|---|---|---|---|---|---|---|
| Model | Curve | Min | Max | Mean | Min | Max | Mean |
| PAT ($\lambda = 25$) | Linear | 6.64±0.09 | 228.58±110.62 | 37.73±19.92 | 75.01±11.77 | 99.32±0.01 | 95.80±2.16 |
| PAT ($\lambda = 25$) | Bézier (Ham.) | 6.64±0.07 | 8.14±0.53 | 7.11±0.10 | 99.12±0.09 | 99.32±0.01 | 99.26±0.01 |
| PAT ($\lambda = 25$) | Bézier (BCE) | 6.65±0.11 | 7.78±0.13 | 7.35±0.06 | 99.18±0.02 | 99.32±0.01 | 99.24±0.01 |
| PAT ($\lambda = 50$) | Linear | 6.47±0.12 | 251.09±102.46 | 43.46±22.46 | 71.57±10.24 | 99.34±0.01 | 95.01±2.39 |
| PAT ($\lambda = 50$) | Bézier (Ham.) | 6.46±0.12 | 8.01±0.40 | 7.00±0.09 | 99.15±0.05 | 99.34±0.01 | 99.28±0.01 |
| PAT ($\lambda = 50$) | Bézier (BCE) | 6.47±0.08 | 7.76±0.12 | 7.35±0.05 | 99.18±0.02 | 99.34±0.01 | 99.24±0.01 |
| PAT ($\lambda = 75$) | Linear | 6.49±0.04 | 345.56±133.60 | 82.49±44.28 | 63.04±13.58 | 99.34±0.00 | 91.13±4.60 |
| PAT ($\lambda = 75$) | Bézier (Ham.) | 6.47±0.05 | 8.22±0.79 | 7.02±0.16 | 99.11±0.12 | 99.34±0.00 | 99.27±0.02 |
| PAT ($\lambda = 75$) | Bézier (BCE) | 6.51±0.04 | 8.12±0.72 | 7.40±0.08 | 99.16±0.05 | 99.34±0.00 | 99.23±0.01 |
| PTT | Linear | 6.70±0.07 | 332.39±116.90 | 107.24±42.20 | 63.67±11.94 | 99.32±0.01 | 87.59±4.31 |
| PTT | Bézier (Ham.) | 6.70±0.07 | 15.33±5.54 | 9.56±1.07 | 97.98±0.94 | 99.32±0.01 | 98.91±0.18 |
| PTT | Bézier (BCE) | 6.69±0.07 | 7.60±0.21 | 7.07±0.06 | 99.21±0.03 | 99.32±0.01 | 99.28±0.01 |

Table 3: Comparison of minimum, maximum, and average hamming loss and true positive rate for models sampled alongside low-energy connecting linear and Bézier curve (three anchor points; minimizing hamming or binary-cross-entropy loss) between PTT and PAT modes. (↑: higher is better; ↓ lower is better)

| | | Hamming loss [$\times 1e^{-3}$] (↓) | | | True positive rate [%] (↑) | | |
|---|---|---|---|---|---|---|---|
| Model | Curve | Min | Max | Mean | Min | Max | Mean |
| $GNN_{PTT} \rightarrow GNN_{PAT}$ | Linear | 6.60±0.13 | 337.23±169.20 | 117.42±76.26 | 63.90±17.66 | 99.33±0.01 | 87.12±8.06 |
| $GNN_{PTT} \rightarrow GNN_{PAT}$ | Bézier (Ham.) | 6.60±0.13 | 11.32±3.25 | 8.06±0.58 | 98.65±0.52 | 99.33±0.01 | 99.13±0.08 |
| $GNN_{PTT} \rightarrow GNN_{PAT}$ | Bézier (BCE) | 6.59±0.12 | 7.75±0.36 | 7.19±0.09 | 99.19±0.03 | 99.33±0.01 | 99.26±0.01 |

Supplemented by the results in Table 2, we find that connecting curves with consistently low-loss values and high true positive rates are existent for models with the same training configurations. However, despite the existence of the excellent connectivity with Bézier curves, the linear connectivity is associated with persistently high losses indicating low mechanistic similarity. Analogous to the functional similarity results in Figure 7, we report increasing losses for higher interpolation values, strengthening the findings in Section 5.3 pointing towards the importance of the interpolation factor despite similar reconstruction performances.

Despite the lack of linear connectivity, given the marginal difference in non-linear connecting curves between all PAT-trained models, all parameter configurations seem to be strongly interlinked with globally well-connected minima. This property also holds for the PTT configuration; however, we were only able to find low-loss connecting curves by using the BCE-loss as the optimization criterion. Optimizing the Hamming-loss, in contrast, yielded substantially worse results, which we cannot explain.

We find similar results demonstrating strong nonlinear connectivity between trained models of PAT and PTT mixtures, indicating a good global connectivity of the modes without noticeable loss barriers that could indicate more pronounced differences in the optimization process (see Table 3). However, the high loss values

---

[9]Trained models using different mechanisms or representations for predictions.

of the linear connectivity (mean$_{PTT \to PAT}$: 117.42±76.26, vs. e.g., mean$_{PTT}$: 107.24±42.2) demonstrate a noticeable impact of mixing the training paradigms on the mechanistic similarity. In addition, we find similar to the connecting curves of PTT minimizing the Hamming-loss noticeable higher loss values compared to the BCE-loss.

**Interpretation:** *While analyzing the global connectivity of loss landscape does not replace the analysis of E2E-differentiability in an entire processing pipeline, we argue that good global connectivity of local loss basins which are interconnected via a low-loss non-linear path is a good initial indication that E2E particle tracking can be efficiently augmented by gradients originated from a downstream reconstruction or analysis loss. Finding a parameter configuration in the low-loss region that minimizes the performance of the downstream task is thus likely effective without getting stuck in local regions, that are separated by high loss barriers. Further, the low mechanistic, representational and functional similarities (see Section 5.3 and 5.4) suggests that a diverse set of reconstruction policies can be reached in those interconnected regions.*

## 6   Discussion and Conclusion

In this work, we propose and present first studies on E2E differentiable neural charged particle tracking using graph neural networks. We integrate combinatorial optimization mechanisms used for generating disconnected track candidates from predicted edge scores directly into the training pipeline by leveraging mechanisms from decision-focused learning. By optimizing the whole track reconstruction pipeline in an E2E fashion, we obtain comparable results to two-step training, minimizing the BCE-loss of raw edge scores. However, we demonstrate the predict-and-track approach can provide gradient information that can be propagated throughout the reconstruction process. While the usage for simply optimizing the reconstruction performance of the network is limited, providing gradient information is highly valuable for various use-cases such as uncertainty propagation, reconstruction pipeline optimization and allows the construction of complex, multistep architectures, potentially allowing to reduce the combinatorial nature of particle tracking by ensuring unique hit assignment.

By examining the global loss landscape, we observe that, despite similar training behaviors leading to globally well-connected minima with comparable reconstruction performance, there are discernible differences in the learned representations leading to substantial prediction instability. This instability is evident across random initializations and is further exacerbated by comparing models optimized for prediction- and task loss, respectively. The observed prediction instability highlights the importance of E2E differentiable solutions, given that the impact of model instability on separate downstream tasks, such as image reconstruction in pCT, is unpredictable and thus especially crucial for safety-critical applications. Therefore, incorporating the functional requirements of downstream tasks through an additional loss term appears to be highly beneficial. Given the strong training performance of the E2E architecture as well as the mostly convex shape of the two-dimensional loss surfaces with globally well-connected minima, the combined optimization of tracking and downstream task loss promises to be feasible and effective. However, as this would have exceeded the scope of this paper, we leave this for future work.

Our analysis demonstrates, beside the similar reconstruction results obtained for different interpolation values and the general understanding that the exact choice of the hyperparameter is non-critical (Vlastelica et al., 2020), a coupling of the choice of $\lambda$ on the prediction stability of the network and an inverse dependency on the convergence speed. Thus, selecting large interpolation constants, improves the informativeness of gradient information, resulting in faster convergence. The gain in convergence speed, however, comes at the cost of decreased robustness. Selecting lower values for lambda is thus especially important in critical applications where a consistency of outputs is desirable or even essential.

Moving forward, we plan to expand the framework by integrating E2E tracking into reconstruction code incorporating auxiliary losses for downstream tasks, especially for the use-case of pCT. Additionally, we will continue extending our studies to explore the out-of-distribution robustness and general robustness to distribution shifts introduced by simulation to reality gaps. Further, we plan extending our work to other potential combinatorial solvers for generating disconnected track candidates from predicted edge scores, as

well as investigating additional network architectures to gain a more profound understanding of the impact of end-to-end optimization on charged particle tracking.

## Members of the Bergen pCT Collaboration

Max Aehle[a], Johan Alme[b], Gergely Gábor Barnaföldi[c], Tea Bodova[b], Vyacheslav Borshchov[d], Anthony van den Brink[e], Mamdouh Chaar[b], Viljar Eikeland[b], Gregory Feofilov[f], Christoph Garth[g], Nicolas R. Gauger[a], Georgi Genov[b], Ola Grøttvik[b], Håvard Helstrup[h], Sergey Igolkin[f], Ralf Keidel[i], Chinorat Kobdaj[j], Tobias Kortus[a], Viktor Leonhardt[g], Shruti Mehendale[b], Raju Ningappa Mulawade[i], Odd Harald Odland[k, b], George O'Neill[b], Gábor Papp[l], Thomas Peitzmann[e], Helge Egil Seime Pettersen[k], Pierluigi Piersimoni[b,m], Maksym Protsenko[d], Max Rauch[b], Attiq Ur Rehman[b], Matthias Richter[n], Dieter Röhrich[b], Joshua Santana[i], Alexander Schilling[i], Joao Seco[o, p], Arnon Songmoolnak[b, j], Ákos Sudár[c, q], Jarle Rambo Sølie[r], Ganesh Tambave[s], Ihor Tymchuk[d], Kjetil Ullaland[b], Monika Varga-Kofarago[c], Boris Wagner[b], RenZheng Xiao[b, v], Shiming Yang[b], Hiroki Yokoyama[e]

a) Chair for Scientific Computing, TU Kaiserslautern, 67663 Kaiserslautern, Germany b) Department of Physics and Technology, University of Bergen, 5007 Bergen, Norway; c) Wigner Research Centre for Physics, Budapest, Hungary; d) Research and Production Enterprise "LTU" (RPELTU), Kharkiv, Ukraine; e) Institute for Subatomic Physics, Utrecht University/Nikhef, Utrecht, Netherlands; f) St. Petersburg University, St. Petersburg, Russia; g) Scientific Visualization Lab, TU Kaiserslautern, 67663 Kaiserslautern, Germany; h) Department of Computer Science, Electrical Engineering and Mathematical Sciences, Western Norway University of Applied Sciences, 5020 Bergen, Norway; i) Center for Technology and Transfer (ZTT), University of Applied Sciences Worms, Worms, Germany; j) Institute of Science, Suranaree University of Technology, Nakhon Ratchasima, Thailand; k) Department of Oncology and Medical Physics, Haukeland University Hospital, 5021 Bergen, Norway; l) Institute for Physics, Eötvös Loránd University, 1/A Pázmány P. Sétány, H-1117 Budapest, Hungary; m) UniCamillus – Saint Camillus International University of Health Sciences, Rome, Italy; n) Department of Physics, University of Oslo, 0371 Oslo, Norway; o) Department of Biomedical Physics in Radiation Oncology, DKFZ—German Cancer Research Center, Heidelberg, Germany; p) Department of Physics and Astronomy, Heidelberg University, Heidelberg, Germany; q) Budapest University of Technology and Economics, Budapest, Hungary; r) Department of Diagnostic Physics, Division of Radiology and Nuclear Medicine, Oslo University Hospital, Oslo, Norway; s) Center for Medical and Radiation Physics (CMRP), National Institute of Science Education and Research (NISER), Bhubaneswar, India; t) Biophysics, GSI Helmholtz Center for Heavy Ion Research GmbH, Darmstadt, Germany; u) Department of Medical Physics and Biomedical Engineering, University College London, London, UK; v) College of Mechanical & Power Engineering, China Three Gorges University, Yichang, People's Republic of China

## Acknowledgments

We thank the anonymous reviewers for their thoughtful comments that greatly improved the final version of this manuscript. This work was supported by the German federal state Rhineland-Palatinate (Forschungskolleg SIVERT), the Research Council of Norway (Norges forskningsråd), and the University of Bergen, grant number 250858. The simulations and training were partly executed on the high-performance cluster "Elwetritsch" at the University of Kaiserslautern-Landau, which is part of the "Alliance of High-Performance Computing Rhineland-Palatinate" (AHRP). We kindly acknowledge the support of the regional university computing center (RHRK). Tobias Kortus and Nicolas Gauger gratefully acknowledge the funding of the German National High-Performance Computing (NHR) association for the Center NHR South-West. The ALPIDE chip was developed by the ALICE collaboration at CERN.

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

## A    Edge Filter Results

In Section 4.1 we mentioned the necessity of performing edge filtering, minimizing the size of both the hit- and line graph representations, improving reconstruction speeds and minimizing memory requirements during training and inference. Selecting sufficient thresholds, given the stochastic nature of particle scattering in material, requires a trade-off between reduced graph size and the number of illegitimately removed true edges. Figure 8 shows the fraction of removed number of edges in the graph opposed to the fraction of removed true edges, generated for 100 graphs of the train dataset selected uniformly over all phantom thicknesses. Based on the results in Figure 8 we select the thresholds $\theta_d = 400$ mrad for the edges in the calorimeter layer and $\theta_t = 200$ mrad for all edges contained in the tracking layer, significantly reducing the number of total edges, while limiting the impact on the final reconstruction quality to a minimum. Error rates for both $\theta_t$ and $\theta_d$ are presented in Figure 8.

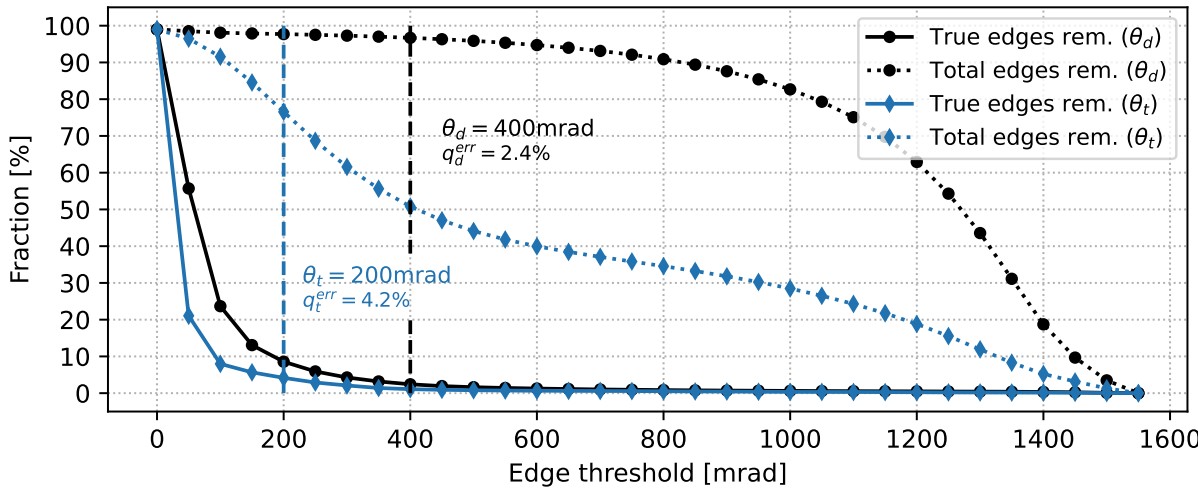

Figure 8: Fraction of total and true edges removed from $\mathcal{G}_H$ given the edge filter $\theta_d$ and $\theta_t$. Measured for 100 sampled graphs from the training dataset with $100p^+/F$ and 100, 150 and 200 mm water phantoms.

## B    Hyperparameters

Table 4: Selected hyperparameters used in the studies in Section 5. All hyperparameters, except $\lambda$ are kept consistent across training runs for PAT and PTT. $\lambda$ is selected based on the value ranges in (Vlastelica et al., 2020; Rolínek et al., 2020b) as we consider it as one of the main parameters for E2E training.

| Name | Value | Description |
|---|---|---|
| $\theta_d$ | 400 mrad | Filter threshold for edges in the calorimeter layers. |
| $\theta_t$ | 200 mrad | Filter threshold for edges in the tracking layer. |
| $n_{hidden}(R1)$ | 3 | Number of network layers |
| $n_{hidden}(O)$ | 3 | Number of network layers |
| $n_{hidden}(R2)$ | 3 | Number of network layers |
| $d_{hidden}$ | 16 | Hidden size of the network layer |
| scaling | 0.001 | Scaling factor of the sinusoidal encoding (see Section 4.1) |
| batch size | 32 | Number of reconstructed graphs in a single batch |
| lr | 1e-3 | Learning rate used for parameter updates with RMSProp |
| $\lambda$ | $\{25, 50, 75\}$ | Interpolation factor for blackbox differentiation of LSA layer |

## C    Reconstructed Particle Tracks

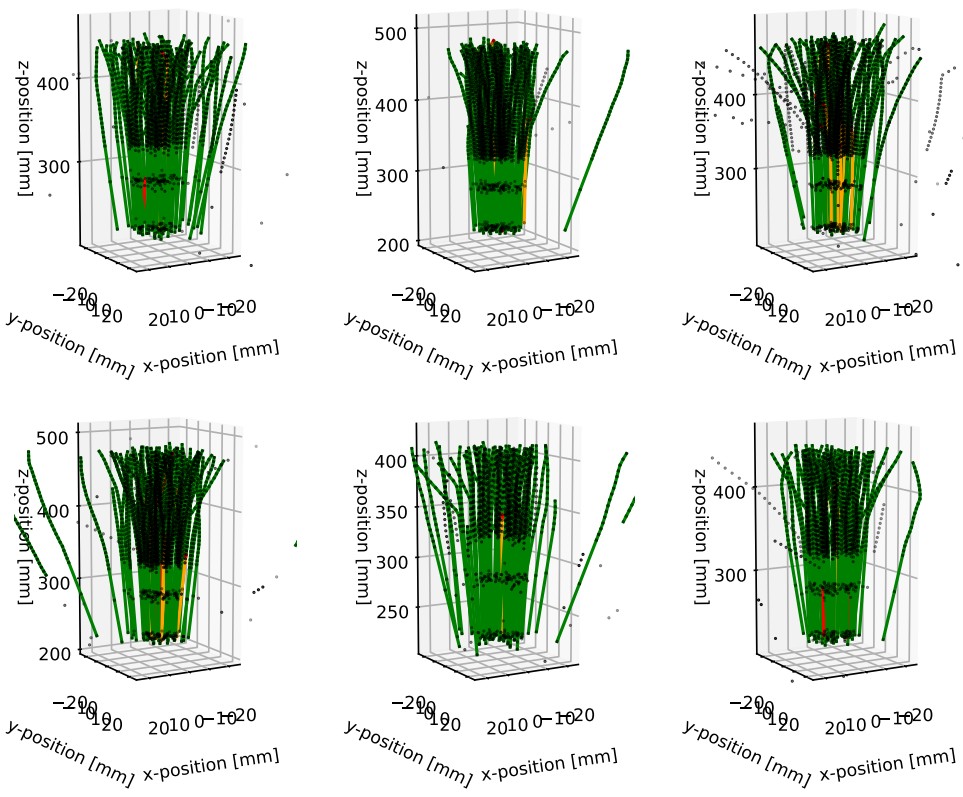

Figure 9: Reconstructed particle tracks generated for multiple readout frames with 100mm water phantom and 100 $p^+/F$ using PAT ($\lambda = 25$) model. Marked in green are correctly reconstructed track segments, marked in red are incorrectly reconstructed track segments and marked in orange are correct reconstructions (same particle ID) following the wrong primary particle.

## D    Ablation Study: Hit-graph and Line-graph Representations

While demonstrating good training performance, utilizing a line-graph representation for particle tracking comes at the cost of increased graph size and thus computational overhead over the original hit-graph representation. To verify the necessity and usefulness of this transformation over a default hit graph parameterization, this section provides additional ablation results comparing hit- and line-graph representations in terms of convergence behavior, reconstruction performance and inference speed.

Reference results utilizing a hit-graphs are generated for all model configurations outlined in Section 5 by replacing the modified architecture with the original interaction-network implementation (Battaglia et al., 2016; 2018) used in DeZoort et al. (2021). The assignment operation, described in Section 4.2 is left as is. We parameterize both vertex- ($\boldsymbol{v}_i$) and edge features ($\boldsymbol{e}_{ij}$) following previous work by (Kortus et al., 2023), using energy deposition and global position ($\Delta E, x, y, z$) and a description of the edge in terms of spherical coordinates ($r, \theta, \phi$) orthogonal to the detector plane respectively (see Figure 2). Table 5 summarizes the average node and edge count ($\pm$ standard deviation) for each evaluated configuration. For every interaction-network configuration operating on a hit-graph, the same hyperparameters from Appendix B are utilized except for the hidden dimension $d_{hidden}$, which is increased to 64. While we found that the

interaction network operating on the hit-graph benefits from the increased hidden dimensionality, we find similar to DeZoort et al. (2021) that the model saturates around the selected value and further increase in parameter count does not yield any noteworthy improvement.

Table 5: Average node- and edge count of hit- and line-graph representations ($\pm$ standard deviation), for simulated data with 100, 150 and 200 m water phantoms and 5, 100 and 150 primaries per frame ($p^+/F$).

| $p^+/F$ | Repr. | 100 mm Water Phantom | | 150 mm Water Phantom | | 200 mm Water Phantom | |
|---|---|---|---|---|---|---|---|
| | | #nodes | #edges | #nodes | #edges | #nodes | #edges |
| 50 | $\mathcal{G}_H$ | 1337±54 | 4311±497 | 1074±38 | 3079±397 | 815±28 | 2235±284 |
| | $\mathcal{G}_L$ | 4311±497 | 68328±16107 | 3079±397 | 39584±10795 | 2235±284 | 24815±7541 |
| 100 | $\mathcal{G}_H$ | 2675±76 | 14837±1350 | 2148±53 | 10397±1015 | 1628±40 | 7514±803 |
| | $\mathcal{G}_L$ | 14837±1350 | 493910±86354 | 10397±1015 | 279974±53359 | 7514±803 | 172926±41226 |
| 150 | $\mathcal{G}_H$ | 4015±80 | 31532±2413 | 3224±62 | 21906±1836 | 2445±46 | 15770±1400 |
| | $\mathcal{G}_L$ | 31532±2413 | 1611761±232139 | 21906±1836 | 908953±153476 | 15770±1400 | 552870±96190 |

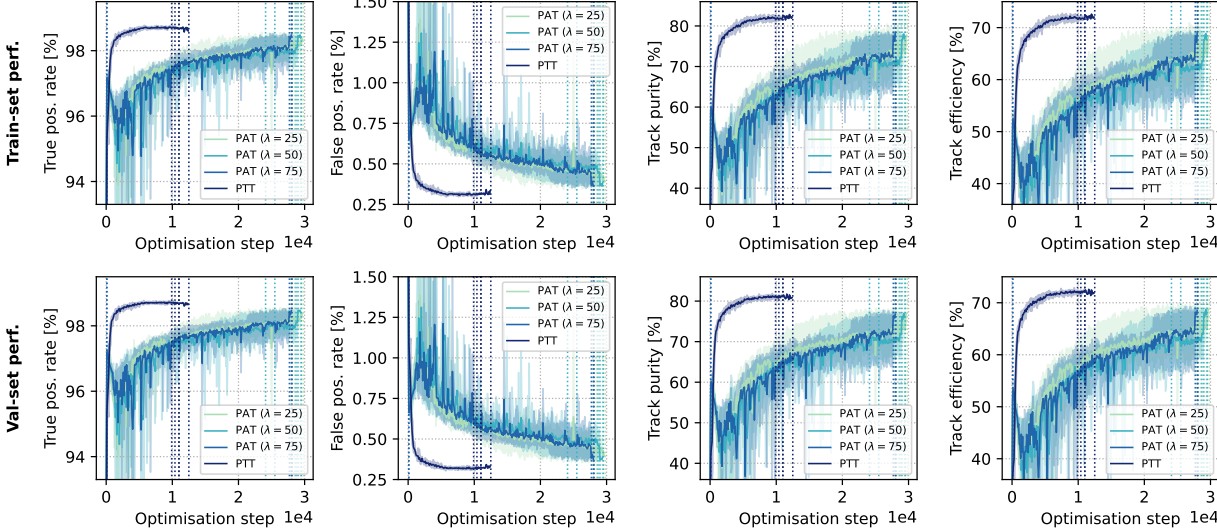

Figure 10: True positive and false positive rate of assignments, as well as purity and efficiency of track reconstruction, evaluated for PAT and PTT with hit-graph representation on validation set as a function of optimisation steps.

**Training performance:** Figure 10 visualizes analogous to Figure 4 the performance metrics on training and validation set as a function of training steps. We find that restricting the readout representation to a hit-graph severely impedes training performance. Across all model configurations, we find a significant drop across all measured quantities. Especially, end-to-end optimization strategies suffer from the reduced information density encoded in the line-graph representation. This results both in decreased convergence speed and stability, as well as reduced reconstruction quality.

**Reconstruction performance:** Table D further confirms the previous finding, comparing the reconstruction quality of the hit-graph baseline ($\mathcal{G}_H$) with the line graph ($\mathcal{G}_L$) results, first presented in Table 5.1. We find a substantial drop in reconstruction purity and efficiency across all tested phantom geometries and particle densities when the line graph transformation is omitted. This effect is especially pronounced for the end-to-end trained baseline, confirming the necessity of the line-graph transformation and the importance of the direct parameterization of segment-pair features, capturing the scattering behavior of particles, for

Table 6: Reconstruction performance for hit- and line-graph representations, measured as purity $p$ and efficiency $\epsilon$ for water phantoms of 100, 150 and 200 mm thickness and 50, 100 and 150 primaries per frame $p^+/F$. The results for $\mathcal{G}_H$ were previously presented in Table 5.1.

(↑: higher is better; ↓ lower is better)

| $p^+/F$ | Repr. | Algorithm | 100 mm Water Phantom | | 150 mm Water Phantom | | 200 mm Water Phantom | |
|---|---|---|---|---|---|---|---|---|
| | | | $p$ [%] (↑) | $\epsilon$ [%] (↑) | $p$ [%] (↑) | $\epsilon$ [%] (↑) | $p$ [%] (↑) | $\epsilon$ [%] (↑) |
| 50 | $\mathcal{G}_H$ | GNN$_{PTT}$ (BCE. edge score) | 81.9±5.8 | 71.6±5.3 | 89.9±0.8 | 80.9±0.7 | 92.3±0.1 | 84.8±0.1 |
| | | GNN$_{PAT}$ ($\lambda = 25.0$) | 73.5±4.1 | 64.0±3.8 | 83.9±2.0 | 75.6±1.8 | 88.8±2.1 | 81.7±2.0 |
| | | GNN$_{PAT}$ ($\lambda = 50.0$) | 74.9±5.7 | 65.2±5.6 | 83.0±2.8 | 74.7±2.6 | 87.5±2.5 | 80.4±2.3 |
| | | GNN$_{PAT}$ ($\lambda = 75.0$) | 74.8±3.1 | 65.3±2.7 | 81.3±3.9 | 73.1±3.5 | 84.9±4.5 | 77.9±4.3 |
| | $\mathcal{G}_L$ | GNN$_{PTT}$ (BCE. edge score) | 94.9±0.1 | 83.8±0.0 | 96.2±0.2 | 86.9±0.2 | 96.4±0.0 | 88.7±0.1 |
| | | GNN$_{PAT}$ ($\lambda = 25.0$) | 94.9±0.1 | 83.8±0.1 | 96.2±0.1 | 87.0±0.1 | 96.5±0.1 | 88.9±0.1 |
| | | GNN$_{PAT}$ ($\lambda = 50.0$) | 95.0±0.2 | 83.9±0.2 | 96.3±0.2 | 87.1±0.2 | 96.5±0.1 | 89.0±0.1 |
| | | GNN$_{PAT}$ ($\lambda = 75.0$) | 95.0±0.2 | 83.9±0.2 | 96.3±0.1 | 87.1±0.1 | 96.5±0.0 | 89.0±0.0 |
| 100 | $\mathcal{G}_H$ | GNN$_{PTT}$ (BCE. edge score) | 66.2±7.8 | 56.0±6.9 | 79.8±1.4 | 71.1±1.3 | 83.8±0.1 | 76.2±0.1 |
| | | GNN$_{PAT}$ ($\lambda = 25.0$) | 53.5±3.8 | 45.2±3.5 | 70.3±2.6 | 62.7±2.4 | 77.0±3.3 | 70.2±3.1 |
| | | GNN$_{PAT}$ ($\lambda = 50.0$) | 55.7±6.8 | 46.8±6.4 | 70.1±3.7 | 62.5±3.4 | 75.2±3.9 | 68.5±3.6 |
| | | GNN$_{PAT}$ ($\lambda = 75.0$) | 54.9±3.3 | 46.4±2.9 | 66.5±5.5 | 59.2±5.0 | 71.4±6.7 | 64.9±6.2 |
| | $\mathcal{G}_L$ | GNN$_{PTT}$ (BCE. edge score) | 87.4±0.3 | 75.2±0.2 | 91.9±0.3 | 82.4±0.3 | 91.7±0.2 | 83.8±0.1 |
| | | GNN$_{PAT}$ ($\lambda = 25.0$) | 87.3±0.2 | 75.0±0.2 | 91.7±0.2 | 82.1±0.3 | 92.2±0.2 | 84.4±0.2 |
| | | GNN$_{PAT}$ ($\lambda = 50.0$) | 87.4±0.3 | 75.1±0.2 | 92.0±0.1 | 82.4±0.2 | 92.5±0.1 | 84.6±0.1 |
| | | GNN$_{PAT}$ ($\lambda = 75.0$) | 87.4±0.2 | 75.1±0.2 | 91.9±0.1 | 82.4±0.1 | 92.3±0.2 | 84.4±0.2 |
| 150 | $\mathcal{G}_H$ | GNN$_{PTT}$ (BCE. edge score) | 52.6±7.8 | 43.1±6.7 | 69.2±2.0 | 60.8±1.8 | 75.8±0.4 | 68.7±0.4 |
| | | GNN$_{PAT}$ ($\lambda = 25.0$) | 40.0±3.4 | 32.7±3.1 | 57.1±2.8 | 50.2±2.5 | 67.1±3.8 | 61.1±3.5 |
| | | GNN$_{PAT}$ ($\lambda = 50.0$) | 42.0±5.7 | 34.2±5.3 | 56.9±4.0 | 50.1±3.6 | 65.6±4.5 | 59.7±4.2 |
| | | GNN$_{PAT}$ ($\lambda = 75.0$) | 41.0±2.8 | 33.6±2.5 | 53.5±5.4 | 47.1±4.8 | 61.5±7.5 | 55.8±6.9 |
| | $\mathcal{G}_L$ | GNN$_{PTT}$ (BCE. edge score) | 77.5±0.4 | 65.0±0.3 | 84.9±0.3 | 75.3±0.3 | 87.2±0.2 | 79.6±0.2 |
| | | GNN$_{PAT}$ ($\lambda = 25.0$) | 76.7±0.2 | 64.3±0.2 | 84.8±0.3 | 75.1±0.3 | 87.6±0.3 | 80.1±0.3 |
| | | GNN$_{PAT}$ ($\lambda = 50.0$) | 76.8±0.3 | 64.4±0.3 | 85.1±0.2 | 75.5±0.2 | 88.1±0.4 | 80.6±0.4 |
| | | GNN$_{PAT}$ ($\lambda = 75.0$) | 76.6±0.1 | 64.3±0.1 | 85.0±0.2 | 75.4±0.2 | 87.8±0.2 | 80.4±0.2 |

both two-stage and end-to-end approaches. In line with (Kortus et al., 2023), highlighting the importance of positional encodings for segment pairs, we specifically credit the significant gain in performance to the inductive bias, introduced by the encodings, simplifying learning the underlying interaction mechanisms defined by multiple Coulomb scattering (see. Section 3.1). This effect further enables the use of smaller models, enhancing parameter efficiency compared to networks that operate on the hit-graph parameterization.

**Computational complexity:** Figure 11 outlines the estimated floating-point operations (FLOP) of graph and line-graph interaction networks, operating on various phantom and density configurations[10]. We provide averaged results obtained over all readout frames for each given configuration.

Ultimately, Figure 12 presents and compares the average reconstruction times of readout frames for both hit-graph and line-graph representation to quantify the runtime penalty of line-graph representations. Both runtimes for the interaction-network prediction (including the line-graph transform) and total reconstruction are presented for various water phantom geometries and particle densities. All results are generated on an NVIDIA A100 GPU for all readout frames in the test set, with 10 evaluations for each readout frame.

We find across all configurations that the line-graph interaction-network noticeably impedes the runtime of the predictive phase, both in terms of FLOP and runtimes, due to the significantly increased and denser connectivity of the graph (see Table 5). Yet, the overall impact on the total reconstruction runtime is manageable for the defined scope, outweighing the increase in runtime by the significant gain in reconstruction quality.

---

[10]All FLOP estimates were calculated using the `FlopCountAnalysis` tool provided as a part of `fvcore` (FAIR, 2019).

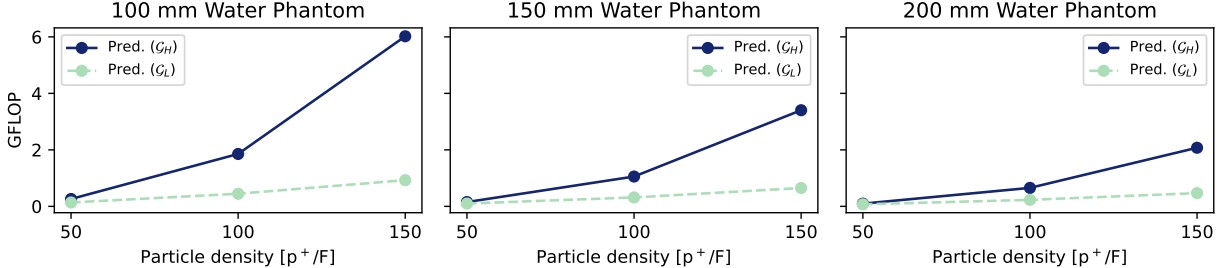

Figure 11: Estimated number of floating-point operations (FLOP) for hit-graph and line-graph interaction networks, evaluated for readout frames of various phantom geometries and particle densities.

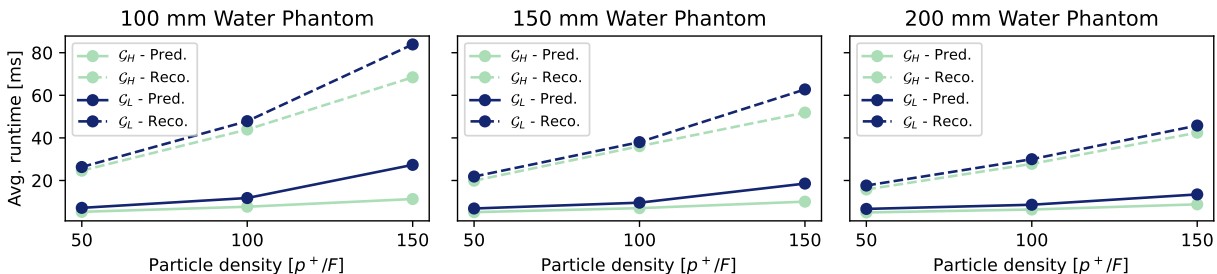

Figure 12: Average runtime of edge-classification charged particle tracking (predictive stage and full reconstruction.) operating on hit-graph ($\mathcal{G}_H$) and line-graph ($\mathcal{G}_L$) representation for readout frames of various water phantom geometries and particle densities.

# E   Implementation Details for Loss Landscape Evaluations in Section 5.2

This section is intended to provide additional implementation details, specifying the implementation and evaluation of the approaches for evaluating the loss landscape in Section 5.2 and all its subsections.

**General remarks:**   As all experiments require a significant amount of network evaluations, often including tens or hundreds of iterations over the data in the training set, we select for Sections 5.2.2,5.2.1,5.3 and 5.4 a subset of 100 uniformly selected graphs from the train set to remain with feasible runtimes, while still providing enough data.

**Loss surfaces:**   We generate the loss surfaces following the implementation details described in Yang et al. (2021). This includes especially the usage of fixed batch normalization parameters across all sampled $\alpha, \beta$ configurations. We generate the eigenvectors of the hessian used as the spanning vectors for the loss landscape using an implementation based on the PyHESSIAN framework (Yao et al., 2020), adapted for our use case. All loss surfaces are generated using 50 x 50 parameter values and parallelized on a single machine with multiple GPUs using the Ray software framework (Moritz et al., 2018).

**Representational and functional similarities**   Our implementation for quantifying representational and functional similarities matches the description of (Nguyen et al., 2020) and (Klabunde & Lemmerich, 2023). However, given the large cardinality and connectivity of the line graph representations, we sample each iteration of the CKA algorithm only a subset of $80 \times 1024$ features to remain with feasible runtimes. Each combination of networks is evaluated as individual jobs using the SLURM resource manager on the Elwetritsch HPC cluster of the University of Kaiserslautern-Landau.

**Mode connectivity:** We closely follow the implementation for generating connecting, curves provided by (Garipov et al., 2018). However, instead of updating the batch normalization parameters for every t, we follow a similar approach used for generating the loss landscapes in Yang et al. (2021). We use constant parameters determined by the first curve anchor, providing us with reasonable normalization values. Each curve is optimized using 1000 iterations with a batch size of 8 graphs. We found that increasing the training iterations resulted with worse connectivity results for PTT-based combinations, minimizing the hamming loss. We use the same parallelization strategy as used for calculating representational and functional similarities.

## F    Reproducibility

We analyze in this work several trained models as well as results based on various evaluation mechanisms, each requiring a significant amount of computing resources. We thus provide for transparency all trained models together with the evaluation results and data under `https://doi.org/10.5281/zenodo.12759188`. Additionally, we provide all the source code for generating the tables and figures used throughout this work, which can be re-run without regenerating any of the required result data. All source code that supports the findings of this study will be openly available open source after paper acceptance under `https://github.com/SIVERT-pCT/e2e-tracking`.

## G    Statistical Testing for Similarity Scores

Given the seemingly increasing predictive instability of the tracking network with increasing interpolation factor $\lambda$, we suspect a dependency between those two dependencies. We verify this hypothesis, using Welch's t-test, demonstrating that the prediction instability decreases with smaller $\lambda$ values (see. Figure 13).

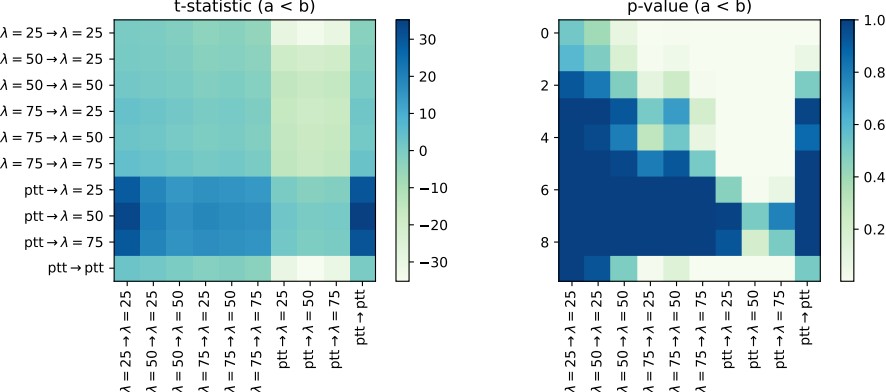

Figure 13: p-values and t-statistic generated for functional similarity values of various model combinations under the hypothesis a (x-axis) < b (y-axis).

