# OpenReview forum: "Exploring End-to-end Differentiable Neural Charged Particle Tracking – A Loss Landscape Perspective"
_TMLR — Accepted by TMLR_

### Review · Reviewer_m8YL · 2024-10-04

**Summary Of Contributions:**

This paper examines charged particle tracking through a graph neural network approach. There are a number of pieces in the work
- End to End optimization with a fully differentiable setup (authors claim that they are the first to do this)
- Linear assignment to discover tracks (differentiable)
- Loss landscape analysis

The loss landscape analysis uses ideas from existing works - such as mode connectivity, and a similarity score between networks.
Favourable evaluations are shown against two baseline methods - Track follower [2] and an RL based approach [2]. They also look at network similarities across parameters and network connectivity across modes.

[1] Track follower (Petterson et al 2020)
[2] RL based approach (Kortus et al 2023)

**Audience:**

Yes

**Claims And Evidence:**

Yes

**Requested Changes:**

Please address my comments in the previous section. I would hope that we can add more context on the problem itself so as to make it a bit more accessible to people who aren't familiar with the area.

Could you also shed some light on what the loss landscape analysis accomplishes? In the fascinating paper [2], for instance, they propose an ensembling idea from this insight - that we can connect modes through Bezier curves. I do see that the results bear out here too, but it somehow feels as though I am missing the point as to what the final word is on the loss analysis in this paper.

[3] Loss Surfaces, Mode Connectivity, and Fast Ensembling of DNNs  https://arxiv.org/pdf/1802.10026

**Strengths And Weaknesses:**

+ The ideas used in the paper are very interesting to dig into and make for elegant reading. I was quite happy to read through the related work and learn how these disparate ideas were used in the particle tracking problem of the text.
+ The results bear out in that end to end differentiability is attained by the trick used in the paper to solve a linear assignment or program and it compares well with the two step approach.
- This is not a familiar area to me (I am a computer vision researcher). However, even with that, it was not easy to get intuition into the problem that is being solved. I think the paper should go into more detail on the nuances of charged particle tracking, as to why end to end modelling is not straight forward in this context, and generally give more background on various domain specific matters (e.g. hit graph vs line graph). That being said, please take this comment with the requisite grain of salt brought on by my lack of domain knowledge.
- I found the loss landscape analysis angle quite interesting, but it seems a bit weak to extend it to the paper. What is the final outcome or summary from the loss landscape analysis? Similarly, for the other analyses on similarity, etc.

---

> ### Author Response · Authors · 2024-10-14
> **Reply to Reviewer m8Y**
>
> We thank the reviewer m8YL for reviewing our work and providing constructive and valuable feedback. We hope that we can improve, with the updated manuscript, the overall understandability and conciseness of our work. In the following, we address the reviewer’s remarks:
>
> - **Provide more detail on the nuances of charged particle tracking:** We added further context and explanations on the topic of charged particle tracking throughout the paper. We especially added an updated description of the challenges to Section 3 together with a new figure (Figure 1), visualizing the combinatorial complexity of particle tracking.
> We hope that the additional references to the combinatorial nature of particle tracking, together with the explanations regarding the integration of discrete assignment operations as a necessary post-processing step, further illustrates the difficulty of E2E differentiable solutions. To provide additional context, we added a reference to [1], discussing the limitations of E2E-differentiability for particle tracking.
>
> - **Give more background on various domain specific matters:** We added more details on the difference of hit- and line-graph representations, explaining the advantages and usage of those representations. We are happy to provide additional insights on this topic or other domain-specific matters if necessary. We trustfully rely on your helpful assessment from a non-domain expert perspective.
>
> - **Clearify the final outcome or summary from the loss landscape analysis:** We agree that the manuscript doesn't reflect and discuss our findings of our loss landscape analysis in enough detail. To clarify this matter, we extended Section 1 and added more emphasis on the complexity of reconstruction and analysis software associated with high-energy physics (Introduction + Chapter 3.2). We additionally provide for each technique a paragraph, interpreting the possible implications of the results with a special focus on the E2E-differentiability in component-based processing pipelines.
>
> [1] Aehle et al., Exploration of Differentiability in a Proton Computed Tomography Simulation Framework (https://iopscience.iop.org/article/10.1088/1361-6560/ad0bdd)

---

### Review · Reviewer_Pqe2 · 2025-03-16

**Summary Of Contributions:**

This paper proposes an end-to-end differentiable framework for charged particle tracking by integrating a GNN based edge classification scheme with a combinatorial optimization step formulated as a layer-wise linear assignment problem (LSAP). By incorporating a black-box differentiation technique, the authors are able to approximate gradients through the otherwise non-differentiable assignment operation, thus enabling true end-to-end training. Extensive experiments on simulated proton computed tomography data compared the end-to-end “predict-and-track” (PAT) approach with the traditional “predict-then-track” (PTT) method. The analysis shows that while reconstruction metrics are similar between the two methods, PAT exhibits faster convergence and more favorable loss landscape properties.

**Audience:**

Yes

**Broader Impact Concerns:**

No concerns

**Claims And Evidence:**

Yes

**Requested Changes:**

Two ablation studies are expected:
1. Include ablation studies that explore the impact of different values of $\lambda$ on training stability, convergence speed, and final performance. This will help in assessing the sensitivity of the black-box differentiation approach.
2. Include ablation studies on using original graphs v.s. line graphs to explore the impact on the prediction performance and computation complexity.

**Strengths And Weaknesses:**

Strengths:
+ The paper presents a novel combination of GNN-based edge classification with a differentiable combinatorial solver, effectively bridging the gap between prediction and discrete optimization.
+ The detailed analysis of both local and global loss landscapes provides valuable insights into the optimization behavior and stability of the trained networks.
+ By ensuring end-to-end differentiability, the method lays a strong foundation for integration with downstream tasks due to enhanced gradient flow and stability.

Weaknesses:
- While the end-to-end method shows benefits in training dynamics and loss landscape properties, the improvements in final reconstruction metrics are relatively limited compared to the baseline.
- The approach relies on a black-box differentiation technique that introduces an interpolation parameter $\lambda$, which appears sensitive and requires careful tuning, potentially complicating practical applications.
- Some aspects of the methodology, such as the specifics of the line graph construction and the edge filtering strategy (to mitigate excessive density), are not explained in sufficient detail. It is also unclear why the line graph is chosen to use, in particular using line graphs often introduces a larger and denser graph, which introduces computation overhead. The authors provide some reasons but those sound subjective. It is better to provide an ablation study for the choice of using line graphs.

---

> ### Author Response · Authors · 2025-03-28
> **Reply to Reviewer Pqe2**
>
> We would like to thank reviewer Pqe2 for the critical assessment of our work and the valuable comments improving our work. We have carefully addressed each of your points (marked in the manuscript in blue) and have provided our response below:
> - **The improvements in final reconstruction metrics are relatively limited compared to the baseline:** We understand that the primary performance gain is relatively limited compared to the baseline. Our work is intended as an initial study on the feasibility of end-to-end particle tracking, demonstrating the strong performance of end-to-end optimization despite the high-combinatorial complexity of the task. Additionally, we focus on the characteristic behavior of the loss landscape, guiding future work for both novel architectures and end-to-end integration into full reconstruction code. We updated the manuscript to clarify our intent and character of this work.
> - **Include ablation studies that explore the impact of different values of lambda:** In Sections 5.1, 5.2, 5.3 and 5.4 we present results comparing the end-to-end approach with different values for λ {25, 50, 75}, selected based on reference values in current literature. Here, we intentionally limited the selection not to exceed the overall computational demands. We acknowledge that the results are currently not sufficiently presented and discussed, and thus updated the manuscript by adding additional references to related work and precisely summarizing the obtained findings.
> - **Include ablation studies on using original graphs v.s. line graphs:** We added an in-depth ablation study comparing hit-graph and line-graph representations in Appendix D including (1) convergence behavior, (2) reconstruction performance and (3) computational complexity in terms of average runtimes, to further strengthen our argumentation. We updated Section 4.3 accordingly, adding references to the ablation study.

---

> > ### Comment · Reviewer_Pqe2 · 2025-06-24
> >
> > Thanks for your response and for updating the draft. Would you mind explaining why using line graphs leads to a more efficient model? Line graphs are typically denser and larger, so I’m curious whether the efficiency comes from the fact that the model applied to the line graph is significantly smaller than the baseline GNNs used on the original hit graph. Thanks!

---

> > > ### Author Response · Authors · 2025-06-25
> > > **Reply to Reviewer Pqe2**
> > >
> > > Thank you very much for your follow-up question. We attribute the beneficial performance of the line-graph mainly to the direct representation and parameterization of two edges of the original hit-graph as a single edge-feature of the new line-graph. This condensation of information, directly summarizing scattering information (the main physical interaction mechanism of protons traversing the detector material) provides a strong inductive bias that otherwise needs to be aggregated over multiple GNN messages in the original architecture. [1] further investigates the effectiveness of the use of sinusoidal encodings, demonstrating a significant performance gain over representations that only utilize hit-graph information.
> > >
> > > As a direct consequence, the size of the hit-graph architecture can be reduced while providing strong performance.
> > >
> > > We have updated the sections in the text (marked in green) to clarify our observations and ensure that our points are communicated more clearly.
> > >
> > > [1] T. Kortus, R. Keidel, N. R. Gauger and Bergen pCT Collaboration, "Towards Neural Charged Particle Tracking in Digital Tracking Calorimeters With Reinforcement Learning," in IEEE Transactions on Pattern Analysis and Machine Intelligence, vol. 45, no. 12, pp. 15820-15833, Dec. 2023, doi: 10.1109/TPAMI.2023.3305027.

---

> > > > ### Comment · Reviewer_Pqe2 · 2025-06-25
> > > >
> > > > Thanks for your response. Could you provide the statistics of the hit graph (the node, edge numbers) v.s. the line graph (the node, edge numbers) and their model computation costs, say using GFLOS? Given these numbers, it may be clearer.

---

> > > > > ### Author Response · Authors · 2025-06-26
> > > > > **Reply to Reviewer Pqe2**
> > > > >
> > > > > Thank you for your valuable suggestion. We have updated Appendix D of the manuscript (marked in green) to include the requested statistics for both, node and edge counts, and the computation costs (in GFLOP).

---

> > > > > > ### Comment · Reviewer_Pqe2 · 2025-06-28
> > > > > >
> > > > > > Thanks for the update. I have no further questions.

---

### Review · Reviewer_ERhZ · 2025-05-25

**Summary Of Contributions:**

In this paper, the authors propose an end-to-end training scheme of graph neural networks for particle tracking. They introduce predict-and-track (PAT) and two-step predict-then-track (PTT) approaches. In addition, linear sum assignment is introduced to determine whether the edge should be assigned under the reconstruction policy.  However, end-to-end training of GNN and linear sum assignment is not new in the GNN and segmentation areas.

**Audience:**

Yes

**Claims And Evidence:**

No

**Requested Changes:**

(1) Revision of section 4 according to the comments above.

(2) Clarify the experiments and provide the empirical evidence.

**Strengths And Weaknesses:**

**Strengths:**

 The application of end-to-end training of GNN for particle tracking is interesting.


**Weaknesses:**

(1) The proposed methods in section 4 are not clearly discussed, and many parts are confusing.

- The definition of $\boldsymbol{v}_i$ in Eq.(8) on page 7 is not discussed.

- The cosine similarity $\boldsymbol{s}_{cos}$ is not clear.

- The function $\phi_{O}$ in Eq.(10) on page 9 is not given.

- The authors claim that transforms the hit graph into the line graph at first. However,  why do we need this transformation, and how does this transformation influences the results is not clearly discussed.

- The source node set and target node set in the linear sum assignment problem (LSAP) in Eq.(12) on page 9 are not well explained.

- The authors claim a layer-wise linear sum assignment in Eq.(12). However, how to perform the layer-wise LSA is not clear.   How can the LSA perform end-to-end training?

- The argmin of $y(\mathcal{C})$ above Eq.(14) is not defined.  What is $y$ here and in Eq.(14)?



(2) The experimental evidence is not significant.

In Table 5 on page 30, it shows that the Line graph representation influences the performance a lot.  However, whether the baseline methods in comparison in Table 1 are not clearly discussed.  I am not sure whether the improvement comes from the better  Line graph representation or from the proposed end-to-end training.   The evidence supporting the claim is not clear.

(3) The contribution to the end-to-end training of GNN seems marginal.


**Typos:**

$f2$ in Eq.(5)

---

> ### Author Response · Authors · 2025-05-28
> **Reply to Reviewer ERhZ**
>
> We sincerely thank the reviewer ERhZ for the thoughtful and constructive feedback on our manuscript. We have carefully addressed each of your points (marked in the manuscript in orange) and have provided our response below:
>
> **(1) The proposed methods in section 4 are not clearly discussed, and many parts are confusing:** We appreciate the reviewer’s honest remarks regarding the shortcomings in our methods section. We have addressed each provided bullet point and clarified our descriptions in the text. We have added additional details on the results on hit graph and line graph representations presented in Appendix D and further added an additional reference to [1] (previously only mentioned in Appendix D), first highlighting the importance of encoding segment pairs (edge features of $G_L$) for improved reconstruction performance.
>
> **(2) The experimental evidence is not significant:** Thank you for your valuable feedback regarding the significance of the experimental evidence. We acknowledge that our original manuscript did not outline the details of the ablation study clearly enough to confidently distinguish whether the performance improvements stemmed primarily from the line graph representation or the end-to-end training framework. To address this, we (1) added additional details on the ablation setup and (2) clearly stated the dependency between Table 1 and the ablation results in Table 5. We further expanded the discussion of the results.
>
> **(3) The contribution to the end-to-end training of GNN seems marginal:** Our work aims to demonstrate the feasibility of end-to-end particle tracking, rather than to introduce entirely new techniques for end-to-end training of GNNs. We believe that the strong performance of end-to-end optimization—despite the high combinatorial complexity of the task—along with the distinctive characteristics of the loss landscape, offers valuable insights for a broad research community. We believe that our findings are relevant to any discipline exploring the integration of discrete assignment operations into neural architectures, and help guide future efforts toward both novel architectural designs and full end-to-end integration within reconstruction pipelines.
>
> [1] T. Kortus, R. Keidel, N. R. Gauger and Bergen pCT Collaboration, "Towards Neural Charged Particle Tracking in Digital Tracking Calorimeters With Reinforcement Learning," in IEEE Transactions on Pattern Analysis and Machine Intelligence, vol. 45, no. 12, pp. 15820-15833, Dec. 2023, doi: 10.1109/TPAMI.2023.3305027

---

> > ### Comment · Reviewer_ERhZ · 2025-06-30
> >
> > I appreciate the authors' feedback and detailed revision.   Most of the concerns have been addressed.
> >
> > However, it seems that the Line graph representation indeed influences the convergence speed and stability, as well as the reconstruction quality a lot, as the authors showed in Appendix D.  I would like to know whether the baseline/SOTA methods using Line graph representation can improve or even perform better than the proposed method. It seems that the baselines in comparison are using the hit-graph representation.   This may be helpful to clarify the value and significance of the proposed end-to-end methods.

---

> > > ### Author Response · Authors · 2025-06-30
> > > **Reply to Reviewer ERhZ**
> > >
> > > Thank you for your positive feedback and your follow-up question regarding the baseline algorithms.
> > >
> > > - The conventional track follower in [1] uses a heuristic search to minimize total scattering angle across paths. Switching to a line-graph representation only affects the track follower’s implementation, not its core logic or outcome.
> > > - Method [2] uses segment pair information similar to our line-graph parameterization, without explicitly constructing the full line-graph. Given the sequential nature of RL, [2] can more efficiently query segment pairs based on partially reconstructed candidates.
> > >
> > > We have updated the manuscript (marked in purple, p.12) to further clarify the relationship between [2] and our method and to categorize the baseline accordingly.
> > >
> > > [1] H. E. S. Pettersen, et al. (2020). Proton Tracking Algorithm in a Pixel-Based Range Telescope for Proton Computed Tomography.
> > >
> > > [2] T. Kortus, R. Keidel, N. R. Gauger and Bergen pCT Collaboration, "Towards Neural Charged Particle Tracking in Digital Tracking Calorimeters with Reinforcement Learning," in IEEE Transactions on Pattern Analysis and Machine Intelligence, vol. 45, no. 12, pp. 15820-15833, Dec. 2023, doi: 10.1109/TPAMI.2023.3305027

---

### Decision · Action_Editor_pLNf · 2025-07-09

**Recommendation:** Accept with minor revision

**Additional Comments:**

A main contribution of the paper is to demonstrate the feasibility of an end-to-end training scheme for particle tracking. However, readers may think that end-to-end GNN training is a common approach in related domains. The authors are therefore encouraged to clarify why a straightforward application of end-to-end GNN methods is not directly applicable to this task, and to highlight the specific challenges, such as the combinatorial complexity that their method addresses.

**Audience:**

Yes

**Audience Explanation:**

The paper presents a differentiable framework for particle tracking by combining GNN-based edge classification with a differentiable combinatorial solver, which is typically non-differentiable. This is complemented by a thoughtful loss landscape analysis. Overall, the approach is well-motivated and provides a valuable contribution to the literature.

**Claims And Evidence:**

Yes

**Claims Explanation:**

The paper proposes an end-to-end differentiable GNN framework for charged particle tracking, combining edge classification with a differentiable combinatorial solver. The paper includes a thoughtful loss landscape analysis that provides useful insight into the optimization dynamics and training stability of the proposed method. This aspect of the work was well-executed and adds credibility to the claim that end-to-end optimization offers practical benefits comparable to two-stage approaches, particularly in terms of convergence behavior. This work thus supports the feasibility of end-to-end particle tracking, as it shows that the approach performs reasonably well despite the task's high combinatorial complexity.

---

> ### Author Response · Authors · 2025-07-24
> **Reply to Action Editor pLNf**
>
> Dear Action Editor and Reviewers,
>
> Thank you for your favorable final decision. We sincerely appreciate the time and effort you devoted to reviewing our paper and providing valuable feedback.
>
> We have uploaded the camera-ready version, which incorporates your final suggestions to better contextualize the state of the art and the challenges of end-to-end optimization for charged particle tracking. In particular, we have added two recent references that explore alternative approaches to circumvent optimization over discrete assignment operations. These works were published after our initial submission.